# Understanding patterns of HIV multi-drug resistance through models of temporal and spatial drug heterogeneity

Alison F Feder[1,2]*, Kristin N Harper[3], Chanson J Brumme[4,5], Pleuni S Pennings[6]

[1]Department of Integrative Biology, University of California, Berkeley, Berkeley, United States; [2]Department of Genome Sciences, University of Washington, Seattle, United States; [3]Harper Health and Science Communications, LLC, Seattle, United States; [4]British Columbia Centre for Excellence in HIV/AIDS, Vancouver, Canada; [5]Department of Medicine, University of British Columbia, Vancouver, Canada; [6]Department of Biology, San Francisco State University, San Francisco, United States

**Abstract** Triple-drug therapies have transformed HIV from a fatal condition to a chronic one. These therapies should prevent HIV drug resistance evolution, because one or more drugs suppress any partially resistant viruses. In practice, such therapies drastically reduced, but did not eliminate, resistance evolution. In this article, we reanalyze published data from an evolutionary perspective and demonstrate several intriguing patterns about HIV resistance evolution - resistance evolves (1) even after years on successful therapy, (2) sequentially, often via one mutation at a time and (3) in a partially predictable order. We describe how these observations might emerge under two models of HIV drugs varying in space or time. Despite decades of work in this area, much opportunity remains to create models with realistic parameters for three drugs, and to match model outcomes to resistance rates and genetic patterns from individuals on triple-drug therapy. Further, lessons from HIV may inform other systems.

*For correspondence:
affeder@uw.edu

## Introduction

Better anti-retroviral therapies have drastically improved the lifespan of individuals living with HIV (*Marcus et al., 2020*) by preventing the evolution of drug resistance (*Zazzi et al., 2018*). When the first HIV drugs were introduced, drug resistance evolved in nearly all treated individuals in the first 6 months of treatment (*Larder et al., 1989*), and sometimes within weeks (*Schuurman et al., 1995*). This drug resistance, often encoded by single mutations conferring strong resistance (*Larder and Kemp, 1989*; *Günthard et al., 2019*), led to rebounding viral populations and treatment failure.

Triple-drug therapies, introduced in 1995, were expected to prevent the evolution of drug resistance and subsequent treatment failure (*Hammer et al., 1997*). The reasoning behind this is as follows - each of three drugs individually brings the expected number of viruses produced by a single infected cell (the basic reproductive number, $R_0$) below one, causing the viral population to shrink. Because mutations do not confer resistance across multiple classes of anti-retrovirals (unlike many cases of bacterial resistance), even if a virus gets a mutation conferring resistance to one drug, the two other drugs keep the $R_0$ of that virus below one, and so viruses carrying this mutation cannot increase in number (*Nowak and May, 2000*). This limits the spread of single drug resistance mutations and requires that multiple mutations be present for a virus to replicate successfully in the presence of three drugs and cause population growth. Despite HIV's high mutation rate, the probability of multiple mutations emerging simultaneously in a single round of replication is exceedingly small (*Perelson and Nelson, 1999*). However, while triple-drug combination therapies saved many lives

(*Hogg et al., 1999*; *Walensky et al., 2006*), HIV has continued to evolve drug resistance into the modern day (*Lee et al., 2014*; *Rocheleau et al., 2018*).

How HIV can evolve resistance to triple-drug combination therapies represents an enduring mystery. Advances in our understanding of drug resistance evolution on triple-drug therapy are driven mostly by two types of studies. First, analysis of clinical patient data has revealed which drug resistance mutations occur (*Günthard et al., 2019*), how often resistance evolution happens on different drug combinations (*Lee et al., 2014*; *Rocheleau et al., 2018*), and how patient behavior and metabolism correlate with the probability of virologic failure (*Harrigan et al., 2005*; *Bangsberg et al., 2004*). Second, mathematical models have explored how heterogeneous drug levels within an individual over time (for example, due to drug metabolism between doses or treatment interruptions within single patients) (*Rong et al., 2007*; *Braithwaite et al., 2006*; *Rosenbloom et al., 2012*; *Cadosch et al., 2012*; *Bershteyn and Eckhoff, 2013*) and space (for example, due to variable drug penetrance to different anatomical sites within an individual patient) (*Kepler and Perelson, 1998*; *Moreno-Gamez et al., 2015*; *Sanche et al., 2017*) can allow the evolution of resistance, and make certain treatments more likely to fail. While some of these models are complex, few consider combinations of three drugs and stochastic emergence of resistance (see *Supplementary file 1* for a brief overview of salient models). It is therefore not possible to make quantitative comparisons between models and patient data and test how spatial and temporal variation of drug levels in the body contributes to HIV multi-drug resistance evolution on triple-drug therapies.

While no model will ever be sufficiently detailed to capture all the nuances of virus-host-drug interactions in a human body, a significant gap remains in understanding how multi-drug resistance emerges, as revealed by various facets of clinical trial and cohort study data. Specifically, matching models to mutational patterns in clinical data (i.e. which mutations happen, how often, on what timescales, and in what order) can help us better understand whether these models can explain multi-drug resistance evolution in the real world.

In this article, we describe several features of drug resistance evolution in HIV, as observed from clinical data in the form of viral sequences collected from study participants, that we believe have not received the attention they deserve. We then describe two types of models that are good candidates for explaining the observations. The important feature of these models is that they include heterogeneity of selection pressures within the scale of single individuals - either in time or in space - stemming from variable drug decay after administration or penetration to diverse anatomical sites. We extend pre-existing models of spatial and temporal heterogeneity to incorporate three-drug therapies and show how these models can or cannot reproduce specific observations from clinical data for an early triple-drug combination. We close by suggesting a path forward in expanding promising mathematical models of HIV to understand three-drug combinations and testing the models' abilities to match clinical data across multiple dimensions - for example, not just how often drugs fail, but the types of mutational patterns these failures leave in viral genetic data. We also discuss how we think the lessons from HIV can be applied in other systems.

## Results

### Two motivating examples

We first describe two studies that report particularly detailed data on how drug resistance emerges across many individuals with HIV, *Kempf et al., 2004* and *Hoffmann et al., 2009*. Like most modern day therapies, these studies examine drug combinations pairing two nucleoside reverse transcriptase inhibitors (a so-called NRTI therapy backbone) with a drug from a different class - either a protease inhibitor (PI) or a non-nucleoside reverse transcriptase inhibitor (NNRTI). These studies paint a detailed picture of resistance evolution, in that they record both when resistance emerges and which relevant mutations are present.

#### Kempf study

In a clinical trial (M98-863) conducted internationally, 327 participants were randomized to receive treatment with two NRTIs, 3TC (lamivudine) and D4T (stavudine), and a protease inhibitor, NFV (nelfinavir) (*Kempf et al., 2004*). Patient outcomes from this clinical trial after 2 years are plotted in *Figure 1A*. We highlight three observations. First, many viral populations evolved drug resistance.

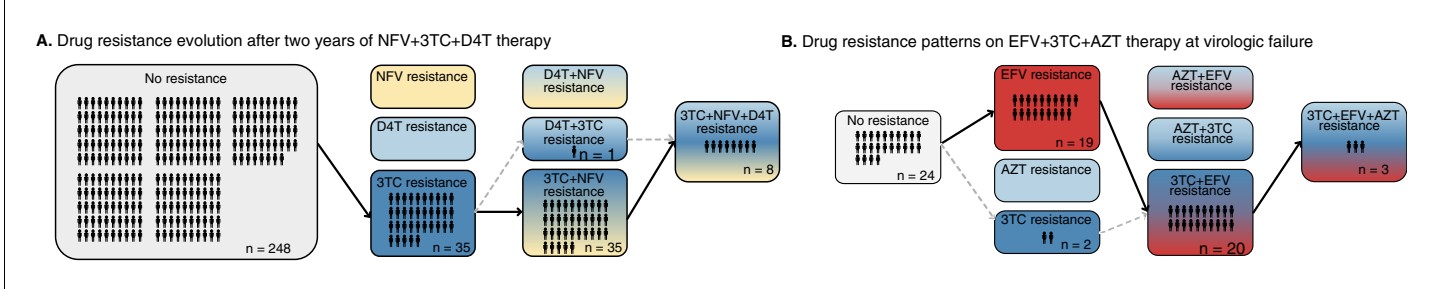

**Figure 1.** Patterns of drug resistance evolution in two studies of triple-drug treated individuals with HIV. (A) Kempf study: Resistance to the drug 3TC emerges first among individuals with HIV taking the triple-drug cocktail NFV + 3TC + D4T (n=327) in *Kempf et al., 2004* in a two year study. NFV is a protease inhibitor, and D4T and 3TC are nucleoside reverse transcriptase inhibitors. (B) Hoffman study: Resistance to the drug EFV emerges first among individuals with HIV taking the triple-drug cocktail EFV + 3TC + AZT (n=68) in *Hoffmann et al., 2009* . Resistance is reported at the time of virologic failure (≤18 months). EFV is a non-nucleoside reverse transcriptase inhibitor (NNRTI), and AZT and 3TC are nucleoside reverse transcriptase inhibitors (NRTIs). Abbreviations: 3TC: lamivudine, AZT: zidovudine, D4T: stavudine, EFV: efavirenz, NFV: nelfinavir. Solid lines (followed from left to right) represent the most frequent progression of drug resistance, whereas dashed lines represent observed minority paths to resistance.

Within the first 2 years of therapy, drug resistance was detected in 79 of the 327 patients (24%). Kaplan-Meier plots in the original paper show that roughly half of these cases occurred in the first year of treatment and half in the second year – indicating that standing genetic variation at the start of treatment was not exclusively responsible for treatment failure. Second, drug-resistant viruses were often not resistant to all three drugs in a treatment. Out of these 79 individuals with resistance, approximately half carried resistance to one drug, half carried resistance to two drugs, and eight patients had resistance to three drugs. Third, drug resistance mutations occurred in a seemingly predictable order. All 35 patients with resistance to one drug had resistance to 3TC and 35 of 36 patients with resistance to two drugs had resistance to 3TC and the PI (NFV). While this is cross-sectional data and we cannot observe the order of failure directly, these data strongly suggest that on 3TC+D4T+NFV treatment, drugs fail in a consistent order: 3TC resistance arises first, then NFV resistance, and finally D4T resistance (see *Figure 1A*).

## Hoffman study

A study in South Africa followed patients treated with a common NNRTI-based regimen containing two NRTI drugs, AZT (zidovudine) and 3TC, and one NNRTI, EFV (efavirenz) (*Hoffmann et al., 2009*). Outcomes for genotyped patients who experienced virologic treatment failure are plotted in *Figure 1B*. Most of the patients experiencing virologic failure who were genotyped (48/68) had drug resistance mutations. However, similar to the Kempf trial, many of these resistant viral populations had only partially resistant genotypes. Of these genotyped patients, 21 had resistance to just one drug, 20 had resistance to two drugs, and 3 had resistance to all three drugs. Again, the partially resistant populations tended to have predictable compositions: 19 of the singly resistant populations were resistant to EFV (the NNRTI) and just two had resistance to 3TC. All patients resistant to two drugs had resistance to EFV and 3TC. These data strongly suggest that on 3TC+AZT+EFV treatment, drugs again fail in a consistent order, although a different order from the PI-based treatment. Here, EFV failed first (*Figure 1B*), and 3TC only second, whereas in *Kempf et al., 2004*, 3TC consistently failed first (*Figure 1A*).

It is well-known that 3TC resistance can evolve extremely quickly when 3TC is used as a monotherapy (*Feder et al., 2021a*). 3TC resistance also evolves first when it is co-administered with AZT (*Picard et al., 2001*; *Larder et al., 1995*) and as we showed in the previous paragraph, it evolves first when co-administered with D4T and the PI NFV. It is therefore somewhat unexpected that when co-administered with an NNRTI, it appears to not be selected unless NNRTI resistance has already appeared.

Several consistent observations emerge from these two studies: (1) significant resistance evolves despite triple-drug treatments. (2) Viruses do not need to be fully resistant to all drugs in a combination in order to spread to detectable levels within individuals, allowing drug resistance to develop

sequentially. (3) Resistance mutations emerge in a partially predictable order. As we discuss below, all three of these patterns are observed much more broadly in larger datasets.

## Three key observations of HIV drug resistance evolution during triple-drug therapy

In this section, we look at bigger, yet less detailed, datasets to determine if the observations from the case studies (resistance evolution is ongoing, sequential, and with a partially predictable order) hold more generally.

### Drug resistance continues to evolve even after years on successful therapy

Despite triple-drug therapies becoming available in high-income countries in the 1990s, drug resistance evolution remained common. Resistance emerged not only in individuals who had started their treatment before 1995 on mono- or double-drug therapies, but also in individuals who started on a triple-drug regimen in the mid-1990s or later. Several studies (*Rocheleau et al., 2018*; *Phillips et al., 2005*) following cohorts of HIV-infected individuals over time found substantial ongoing resistance evolution on both NNRTI-based and PI-based treatments even 5 or more years after starting therapy. In these cohorts, cases of resistance accumulate over time, suggesting that the yearly probability of resistance evolution does not rapidly decline to 0.

To assess how an individual's yearly risk of resistance evolution changes after numerous years on combination therapy, we performed a conditional survival analysis on a previously published cohort of HIV-infected individuals from British Columbia in which all people in the province were automatically entered in a provincial registry and received treatment from a central agency free of charge (*Montaner et al., 2010*; *Rocheleau et al., 2018*). We computed the yearly probability of resistance evolution in each of the first 10 years of treatment, conditional on no prior resistance evolution up to this point, among 2692 individuals treated with three-drug combination therapies commencing between 1996 and 2005. We observe across all four drug categories (NNRTIs, PIs, non-3TC/FTC NRTIs and 3TC/FTC), the yearly conditional probability of resistance stays elevated above 0 even 5 or more years after the start of therapy (*Figure 2A*, *Supplementary file 2*). The conditional survival probabilities are highest in the first year of therapy, and appear to decline thereafter. The point estimates of yearly probability risk are fit well by standard exponential decay for each of the four mutation classes ($r^2 = 0.78, 0.80, 0.82, 0.63$ for 3TC, PI, NNRTI and NRTI resistance, respectively) when excluding the first year, which is expected to have a higher rate of failure due to standing genetic variation (*Pennings, 2012*) or transmitted drug resistance (especially among NNRTIs). The relatively slow rates of decline (*Supplementary file 3*) suggest that the risk of drug resistance evolution on triple-drug therapy remains even in individuals who have been on successful treatment for years. This highlights the importance of ongoing within-patient processes in treatment failure and suggests that resistance patterns are not driven by a subset of individuals who fail therapy quickly.

### Emergence of partially resistant genotypes suggests sequential evolution

Individuals on HIV treatment in high-income countries are tested regularly for viral load. If the viral load is higher than a certain threshold ('viremia' or virologic failure), a viral sample will be sequenced and assessed for resistance mutations (*Department of Health and Human Services. Panel on Clinical Practices for Treatment of HIV Infection. and Henry J. Kaiser Family Foundation. Panel on Clinical Practices for Treatment of HIV Infection., 2000*). Observing sequences with only one or two resistance mutations at this stage suggests that partial resistance can allow growth above the detectable viral load threshold. This is surprising under how we believe triple-drug therapies should work, since a singly-resistant virus should be suppressed by two other drugs and not expand and lead to virologic failure.

In earlier papers, we have mentioned the observation that drug resistance evolves in a step-wise fashion (*Pennings et al., 2014*; *Pennings, 2013*), but did not systematically query this effect across a broad range of triple-drug therapies. To understand if this effect is widespread across treatments, we analyzed a large set of HIV sequences (6717 individuals) from the Stanford HIV Drug Resistance Database. The presented sequences stem from samples at detection of viremia on triple-drug therapy. In these sequences, we identified large-effect drug resistance mutations and quantified the number of mutations per sequence (*Figure 2B*, see methods). While some individuals failed without

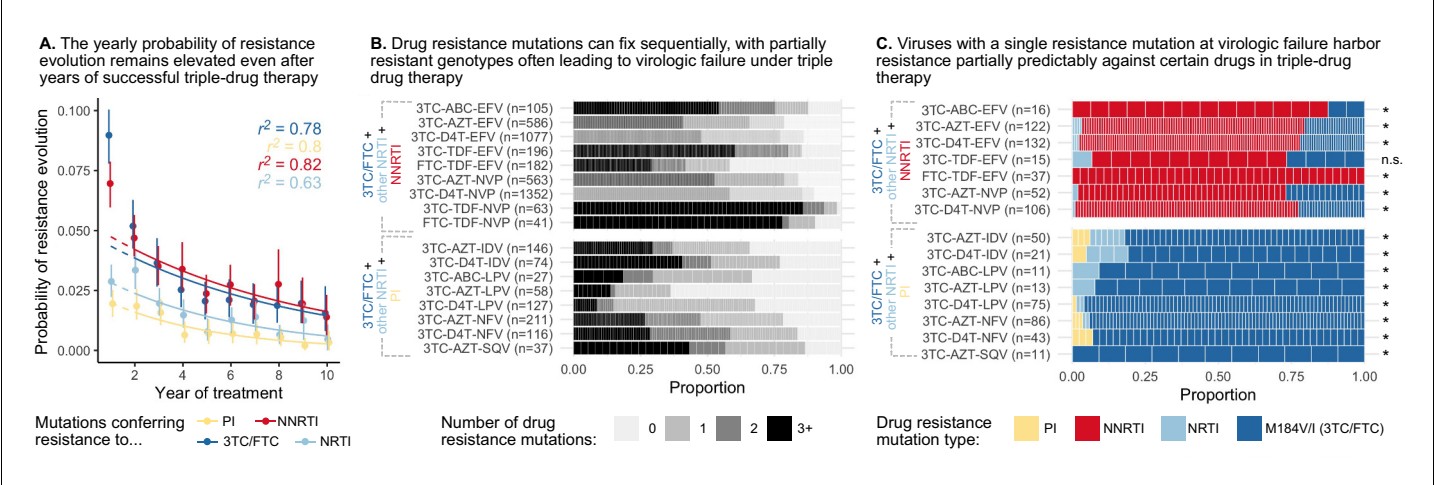

**Figure 2.** Cohort analyses of individuals with HIV reveal drug resistance evolution is ongoing after years on therapy, stepwise and partially predictable. (A) The yearly probability of resistance evolution conditional on no prior resistance evolution in a large cohort of individuals in British Columbia treated with triple drug therapies (*Rocheleau et al., 2018*). Conditional evolution probabilities are plotted separately for resistance to 3TC or FTC (dark blue), NNRTIs (red), NRTIs other than 3TC or FTC (light blue) or PIs (yellow). Drug-category specific fits to an exponential model covering years 2–10 and $r^2$ describing the model fit are also plotted. Bars represent 95% confidence intervals. (B) Across individuals treated with three-drug combination therapies based on two NRTIs paired with an NNRTI (top) or PI (bottom), many sequences are detected with 0, 1, 2, or 3+ drug resistance mutations in a large cohort (data source: *Feder et al., 2016a*). (C) In individuals treated with three-drug combination therapies based on two NRTIs paired with an NNRTI (top) who have exactly one resistance mutation, that mutation most often confers resistance to the NNRTI of the treatment (red), although sometimes resistance to 3TC (dark blue) or the other NRTI (light blue) develops. Among individuals treated with a PI paired with 3TC and a second NRTI who have a single drug resistance mutation, that mutation most often confers resistance to 3TC (dark blue), although sometimes it confers resistance to the other NRTI (light blue) or the PI (yellow). Asterisks mark treatments for which the first mutation confers resistance to the NNRTI (top) or 3TC (bottom) more often than expected under the relative mutation probabilities of the three drugs (binomial test, 5% false discovery rate). (Data source: *Feder et al., 2016a*). Each square in both B and C represents a single individual, and the sample sizes are given next to the treatment names. Abbreviations: abacavir, ABC; indinavir, IDV; lamivudine, 3TC; lopinavir, LPV; nelfinavir, NFV; stavudine, D4T; saquinavir, SQV; zidovudine, AZT.

any detectable drug resistance (likely due to non-adherence), viral populations in many others harbored one, two, three or more resistance mutations. That many sequences harbored only one or two drug resistance mutations suggests that partially resistant virus can replicate in the body and lead to a detectable viral load, even when treated with three drugs. This is consistent with results from *Figure 1* showing that resistance mutations accumulate in a seemingly step-wise fashion. Note, viruses with three or more resistance mutations may still have accumulated resistance mutations one at a time but these intermediate steps may not have been captured by sampling.

## Drug resistance mutations accumulate in a partially predictable order

Possibly the most surprising observation from our two cases studies is that drug resistance mutations emerge in a particular order. Ordering of mutations in HIV has been noted before for different mutations that confer drug resistance to the same drug (*Beerenwinkel et al., 2005a*; *Beerenwinkel et al., 2005b*; *Posada-Céspedes et al., 2020*; *Boucher et al., 1992*; *Molla et al., 1996*). That resistance to one class of drugs tends to precede resistance to another class of drugs has not received much attention in the evolutionary literature. To understand which resistance mutations emerge first on triple-drug therapy, we again examined the Stanford HIV Drug Resistance Database dataset described above and considered only viral populations which harbored a single resistance mutation. For each of those singly resistant populations, we determined the drug to which the mutation conferred resistance (*Figure 2C*). In individuals on PI-based therapies, the singly-resistant viral populations almost all carry a 3TC-resistance conferring mutation, suggesting that resistance to 3TC overwhelmingly occurs before resistance to either the other NRTI or the PI. This ordering of mutations cannot be explained by different mutational target sizes or mutation rates (computed rates given in *Supplementary file 4*). For each triple-drug combination, we computed the sum of the mutation rates for all 3TC resistance mutations versus the total rate for resistance

mutations for any administered drug to get an expected probability that the first resistance mutation to occur should confer resistance to 3TC. We found that for all therapies, resistance to 3TC occurs more often than predicted (p < 0.05, one-sided binomial test with a Benjamini-Hochberg correction and 5% false discovery rate, *Figure 2C*, all test details given in *Supplementary file 5*). This result also cannot be explained by standing genetic variation because the most common 3TC mutation, M184V, incurs a substantial fitness cost (*Paredes et al., 2009*; *Jain et al., 2011*), which should make it unlikely to be present as standing genetic variation (*Simen et al., 2009*). Note, the major 3TC resistance mutation, M184V, also confers some degree of resistance to ABC. Despite this, M184V was counted as a 3TC-resistance conferring mutation when assessing whether 3TC resistance mutations preceded others in the only PI-based, ABC-containing therapy, 3TC+ABC+LPV.

In individuals treated with NNRTI-based therapies, single drug resistance mutations most often conferred resistance to NNRTIs (*Figure 2C*), suggesting NNRTI resistance occurred first. In performing an analogous test comparing the overall mutation rates toward NNRTI resistance versus the resistance mutation rate for any drug in a treatment, we found that in all therapies except one, NNRTI resistance occurred more often than predicted by the relative mutation rates (one-sided binomial test with 5% FDR and Benjamini-Hochberg correction, *Supplementary file 6*). The one triple-drug combination where the null hypothesis was not rejected (3TC-TDF-EFV) also had the fewest number of observations ($n = 15$). Note, we can only comment on the first resistance mutation that is actually observed, but it is also possible that other classes of resistance mutations rise to high frequency before the first mutations identified above but these mutations are then outcompeted due to clonal interference.

Collectively, we have presented three observations that can help guide the construction of models: (1) drug resistance evolution was common on triple-drug therapies in the 1990s and 2000s, and even individuals who had been on treatment for several years were still at risk for resistance evolution; (2) resistance evolves in a sequential manner (first one mutation, then a second, then a third, rather than three mutations at the same time), leading to the observation of partially-resistant viruses; (3) mutations emerge in a partially predictable order that cannot be explained by mutational target size. We next describe two potential explanations for drug resistance evolution on triple-drug therapies and explore how they might produce the patterns described above.

## Two models of resistance evolution due to drug level heterogeneity

Two types of models have been used to explain how drug resistance emerges on combination therapies: models with temporal heterogeneity in drug concentrations (*Rong et al., 2007*; *Braithwaite et al., 2006*; *Rosenbloom et al., 2012*; *Cadosch et al., 2012*; *Bershteyn and Eckhoff, 2013*) and models with spatial heterogeneity in drug concentrations (*Kepler and Perelson, 1998*; *Moreno-Gamez et al., 2015*; *Sanche et al., 2017*). Both models rely on drugs falling below their intended concentrations at certain points in time or space. The literature on temporal heterogeneity is more extensive and includes cohort studies on the adherence-resistance relationship in HIV (*Bangsberg et al., 2004*; *Harrigan et al., 2005*). While nobody doubts that spatial and temporal heterogeneity exist, it is not clear if either or both can explain the observed evolutionary patterns.

Before describing in more detail the spatial and temporal heterogeneity models, we first explain why heterogeneity is crucial - namely, HIV drugs work too well at their administered concentrations for ongoing replication. Even when administered as monotherapy, drugs cause sensitive viral populations to shrink (*Eron et al., 1995*; *Reiss et al., 1988*). As an example, we plot data from a patient treated with AZT monotherapy (*Figure 3A*). Initially, the treated viral population shrinks. Only after 16 weeks, the viral population expands again to detectable levels. Measuring the ability of the drug-resistant and sensitive virus to replicate at different drug concentrations (i.e. the Hill dose-response curve, *Figure 3B*, *Chou, 1976*; *Zhang et al., 2004*; *Sampah et al., 2011*) helps explain the dynamics, and such curves have been measured in cell culture for anti-retroviral-treated HIV (*Sampah et al., 2011*).

At certain intermediate drug concentrations, the resistant mutant can grow ($R_0>1$) and the sensitive strain cannot ($R_0<1$) (*Rosenbloom et al., 2012*, *Figure 3B*). Concentrations within this so-called 'mutant selection window' (which also includes concentrations at which both sensitive and resistant virus have $R_0>1$, and $R_{resistant}>R_{sensitive}$, *Rosenbloom et al., 2012*) select for resistance. The dynamics plotted in *Figure 3A* suggest AZT concentration falls within this window: the population initially shrinks (*Figure 3A*, label 1) suggesting the drug-sensitive genotype must have $R_0<1$ (*Figure 3B*,

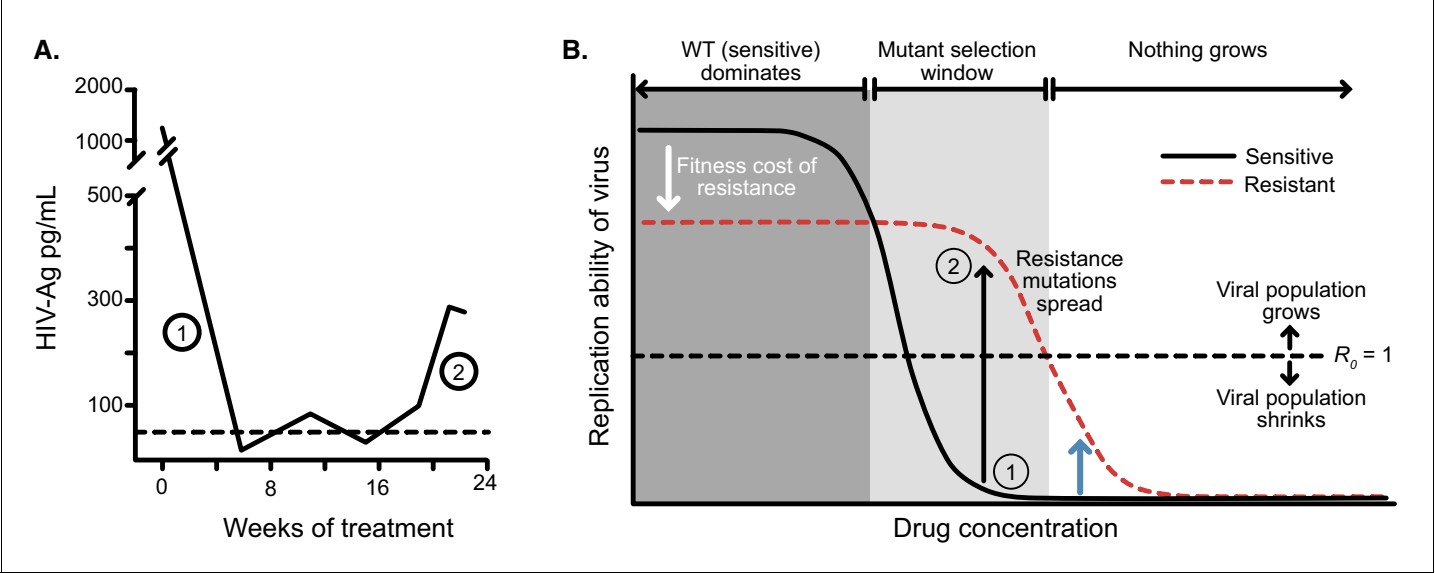

**Figure 3.** Dose response curves help explain the dynamics of treated viral populations. (**A**) HIV antigen levels initially decline when treated with AZT monotherapy (6x200 mg/day) to below a detectable threshold (dotted line). Starting at week 16, the viral population begins to rebound even in the presence of the drug, indicating the evolution of drug resistance. (**A**) Is reprinted with annotations from *Figure 1A* from The Lancet, 1 (8582), Reiss et al, 'Resumption of HIV antigen production during continuous zidovudine treatment', Page 421, Copyright (1988), with permission from Elsevier. It is not covered by the CC-BY 4.0 licence and further reproduction of this panel would need permission from the copyright holder. (**B**) Schematic shows different regimes of viral behavior dependent on the concentration of a particular drug. At high concentrations (right side of the plot), both drug-resistant (dashed red) and drug-sensitive (solid black) viruses are suppressed, with $R_0$ below 1. As drug concentration decreases, a window of concentrations emerges where the drug-resistant virus can grow in population size in the presence of the drug ($R_0 > 1$), but the drug-sensitive genotype cannot. This is termed the 'mutant selection window' (*Drlica, 2003*). At lowest concentrations, both wild-type (WT) and drug-resistant viruses can replicate, but the WT virus out-competes the drug-resistant type because it does not carry the fitness penalty of costly drug resistance mutations. Labels 1 and 2 show how the dynamics of the population in A can be explained by a mutation which behaves like the black arrow in B. If the drug concentration is higher and the resistance mutation only confers a change in $R_0$ according to the blue arrow, the viral population will remain suppressed.

© 1988, Elsevier. is reprinted with annotations from Figure 1A from *Reiss et al., 1988*, Copyright (1988), with permission from Elsevier. It is not covered by the CC-BY 4.0 licence and further reproduction of this panel would need permission from the copyright holder. Figure 3A

label 1), but then begins to grow again (*Figure 3A*, label 2), suggesting that the drug-resistant genotype must be above the $R_0 = 1$ line (*Figure 3B*, label 2). In principle, had AZT dosage been higher (e.g. blue arrow in *Figure 3B*), the resistant virus would be unable to increase in size because its $R_0$ would be below one despite its resistance. Drugs overpowering even resistant virus at high concentrations have been demonstrated empirically in experimental settings (*Sampah et al., 2011*).

Combining three drugs that each bring the viral $R_0$ below 1, should result in an even lower $R_0$ (although how much lower is not straightforward, see *Jilek et al., 2012*). Even if a virus acquires a resistance mutation rescuing its ability to replicate in the presence of one drug, as long as the combination of the three drugs suppresses $R_0$ below 1, that singly resistant virus should not spread. That we see partially resistant viruses reach detectable levels in patients means there must be a time or space where this virus replicates at $R_0 > 1$.

## Temporal heterogeneity

After a patient takes a dose of treatment, the drugs are absorbed and metabolized within the body, resulting in an increase and then a decrease in drug concentration. Ideally, each drug is dosed such that the viral population cannot replicate under normal daily drug fluctuations (*Figure 4A*). However, patient non-adherence and different rates of drug metabolism between individuals can lead to drug concentrations that drop below the required levels over time. This has been widely recognized as a potential route to drug resistance (*Fox et al., 2008*; *Bangsberg et al., 2007*; *Harrigan et al., 2005*; *Chesney et al., 2000*; *Cadosch et al., 2012*).

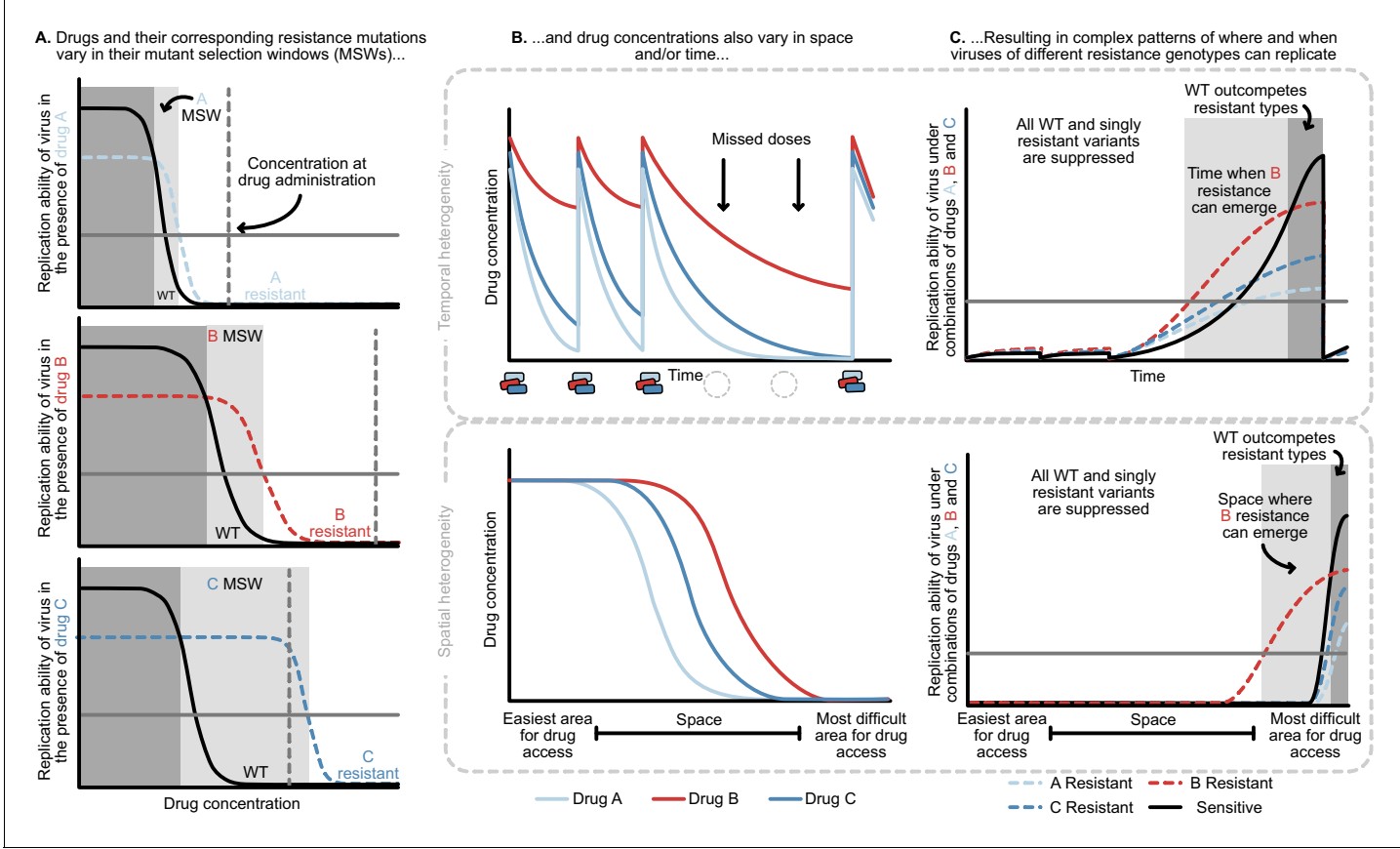

**Figure 4.** Heterogeneity in time or space can allow the spread of drug resistant viruses. (**A**) Different drugs and their associated drug resistance mutations have differently shaped concentration response curves and corresponding mutant selection windows (MSW). Pictured here, drug A (light blue) has the smallest MSW, whereas drug C (dark blue) has the widest MSW. Drug B (red) has an intermediate MSW. (**B**) Variable MSWs also interact with drugs that have heterogeneous concentrations either in time (top) or space (bottom). Missed doses (due to incomplete adherence, plotted on top) can allow only drugs with long half-lives (here, Drug B) to remain present. This can create windows in time (pictured in **C**) which select resistance to only this longest half-life drug. If drug levels are allowed to decline sufficiently low, viral replication of non-resistant types can emerge as well. Drugs that do not reach all of the body (plotted on bottom) can create areas where single drug resistance mutations (pictured in **C**) can replicate. Areas where effectively no drugs reach can create a sanctuary where some viral replication of the wild-type (WT) virus can continue.

The mutant selection window provides a useful lens through which to understand the consequences of missed doses. At drug administration, viral populations experience small $R_0$ values to the right of the mutant selection window and $R_0$ values increase as drug concentrations decay, moving to the left on the x-axis (*Figure 4A*). Drugs with short half-lives move quickly to lower concentrations, while drugs with long half-lives may move slowly through their mutant selection windows. Missed doses may result in drug depletion of quickly decaying drugs while slowly decaying drugs may remain in their mutant selection window. This allows virus resistant to the longest half-life drug to replicate. *Figure 4B and C* show a scenario in which after missed doses, drug B in red with the longest half-life enters the mutant selection window after shorter half-life drugs leave the system. The dynamics clearly depend not just on drug B's mutant selection window, but also the rate at which all drugs are metabolized. To model this situation, it is therefore critical to incorporate the mutant selection windows, relevant drug interactions and drug concentration dynamics for all drugs.

In addition to understanding drug concentrations and the mutant selection windows, several other considerations are important. If the population size is small due to virologic suppression, mutations that confer resistance may not be present, even if the drug conditions would allow them to spread. Second, for the mutant to be successful, the decrease in virus while the patient is taking drugs must be smaller than the increase in virus during the mutant selection window. Patients

missing doses too frequently may not select for resistant virus at all. It is unclear which adherence patterns result in this behavior, and if these same adherence patterns can explain why individuals fail treatment with resistance after 5–10 years on therapy.

We now return to the clinical data and ask whether these patterns may emerge from a model of temporal heterogeneity. Individuals treated with 3TC+D4T+NFV acquire resistance largely in the order of 3TC, then NFV, then D4T. Both NFV and D4T are expected to lose most of their suppression within 24 hr due to short half-lives and, in the case of D4T, low potency (*Rosenbloom et al., 2012*). If a patient misses a dose, 3TC may quickly be the only potent drug present and enter its mutant selection window, allowing a 3TC-resistant genotype to spread. Indeed, we observe that 3TC mutations dominate over PI mutations among single drug resistance mutations in individuals treated with PI therapy and an NRTI backbone containing 3TC (*Figures 1A* and *2C*).

To test whether a quantitative model could create these expected patterns, we extended an existing model of HIV drug resistance evolution (under a variety of adherences) to include combination therapies. Briefly, we simulated 1500 HIV-infected individuals taking 3TC+D4T+NFV (the therapy examined in *Figure 1A*) for a ten-year period and recorded when (if at all) viral populations first became clinically detectable, and which resistance mutations (if any) were present (see Materials and methods for full details). Drugs decay over time according to measured half-lives, and viral fitnesses depend on their genotype and the drug levels currently present. Critically, individuals have an underlying adherence level which controls their probability of missing doses at each administration period.

We found that this model of temporal heterogeneity could explain some but not all features observed in the clinical data under three-drug combination therapies. Consistent with clinical observations, resistance mutations often emerge one mutation at a time and can lead to clinically detectable viral loads (*Figure 5*). This model also produced instances of virologic failure without any resistance mutations, presumably in individuals with particularly poor adherence. Also in agreement with clinical observations, when resistance did evolve, mutations occur in a predictable order and resistance to 3TC always precedes NFV and D4T. In disagreement with clinical observations, we find that resistance overwhelmingly emerges in the first year of treatment when the population size is large, but then falls to quite low levels for the next years. 95.3% of the simulated individuals who did not develop resistance in the first year had suppressed HIV populations for the entire simulated 10-year treatment period. This suggests that under a simple model of non-adherence for 3TC+D4T+NFV, potentially little resistance evolves after initial viral suppression, although we will note in the discussion how more sophisticated models of adherence may more effectively select for resistance.

Above, we simulated a model in which 3TC has the longest half-life in a PI-based drug combination (3TC+D4T+NFV). Unlike PIs, NNRTIs have particularly long half-lives (*Taylor et al., 2007*). Treatment interruptions of a few days are long enough for non-NNRTIs to leave the system, but short enough for NNRTIs to remain present. We would therefore expect NNRTI resistance to occur first, which is consistent with the data presented in *Figures 1A* and *2C*. However, other data are not consistent with this explanation. An observational study conducted as part of the SMART trial on treatment interruptions tried to prevent a period of NNRTI monotherapy when interrupting treatment (so called staggered or switched stops) by 'covering' the tail with additional shorter lived drugs (*Fox et al., 2008*). It turned out that 'covering' the tail did not affect the risk of drug resistance evolution (*Fox et al., 2008*), suggesting that the tail of NNRTI monotherapy may not be the reason NNRTI drug resistance evolves. We (Pennings) have proposed another explanation, namely that an increase in population size during an interruption allows resistance mutations to accumulate as standing genetic variation. When treatment resumes, these mutations increase in frequency due to selection (*Pennings, 2012*). Clearly, these observations demand more investigation into how temporal heterogeneity can generate such patterns, and indeed underscore the challenge of thinking about such observations without a quantitative model.

## Spatial heterogeneity

HIV drugs do not penetrate throughout the body uniformly and penetration profiles differ among drugs. As a result, even in triple-drug-treated individuals, parts of the body (e.g. brain, lymph nodes) may experience monotherapy (*Else et al., 2011*; *Letendre et al., 2008*; *Decloedt et al., 2015*; *Fletcher et al., 2014*; *Thompson et al., 2019*). Under a model of spatial heterogeneity, the concentrations that allow replication and expansion of drug-resistant virus (i.e. the mutant selection window

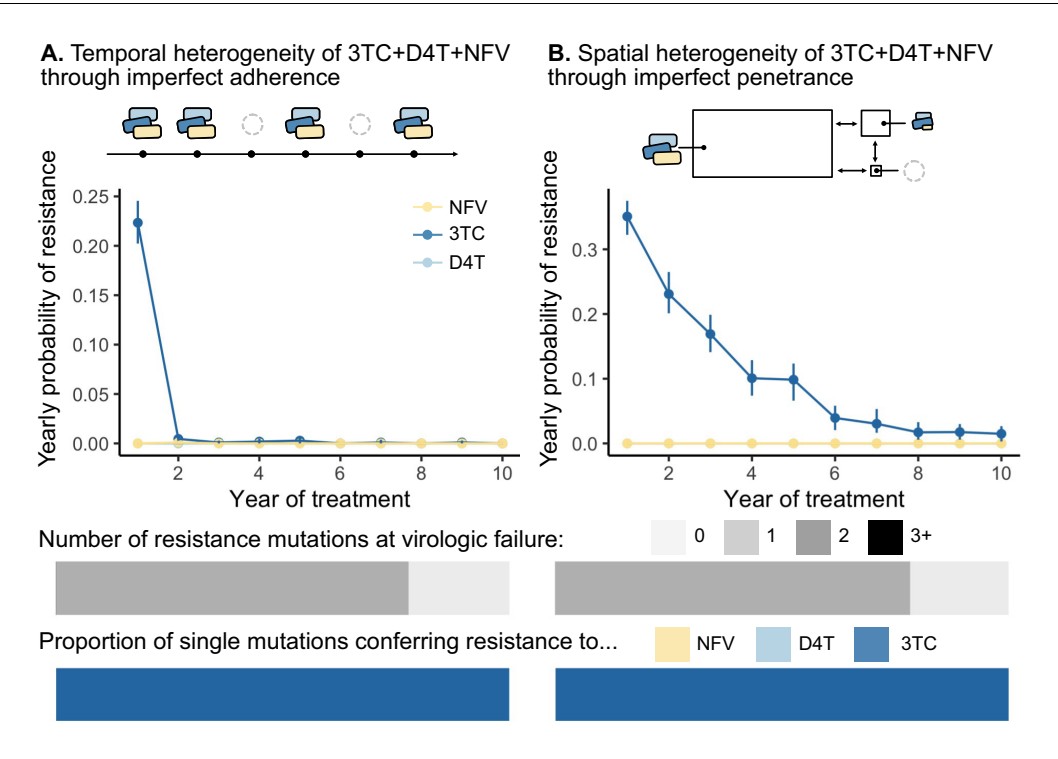

**Figure 5.** Models of spatial and temporal heterogeneity of drug levels explain certain evolutionary patterns observed in clinical data. Using parameterized models of (A) temporal and (B) spatial heterogeneity of drug levels stemming from imperfect adherence and imperfect penetrance, respectively, we can assess the presence or absence of patterns found in clinical data. First, we plot the yearly probability of resistance evolution conditional on no resistance to this point for 3TC, D4T and NFV for each year of treatment. Bars represent 95% confidence intervals. Second, we plot the proportion of individuals with 0, 1, 2 or 3 resistance mutations at the time of virologic failure ($n = 446$ under temporal heterogeneity and $n = 1121$ under spatial heterogeneity). Third, we plot the proportion of individuals with resistance to NFV, D4T, or 3TC conditional on having a single resistance mutation at the time of virologic failure ($n = 346$ under temporal heterogeneity and $n = 878$ under spatial heterogeneity). Note, the cartoon model of spatial heterogeneity in drug levels does not show the compartment sizes to scale.

The online version of this article includes the following figure supplement(s) for figure 5:

**Figure supplement 1.** Clinical endpoint timing covaries with plasma population size in model of spatial heterogeneity of 3TC+D4T+NFV.

**Figure supplement 2.** Yearly probability of resistance evolution under a model of spatial heterogeneity of 3TC+D4T+NFV varies little over time when the population size range is narrow.

of each drug) are distributed in physical space, as opposed to time. These concentration profiles may be largely overlapping and nesting (as illustrated in *Figure 4B*) or drugs may have semi-independent penetrance profiles, wherein some drugs reach certain parts of the body effectively, and other drugs reach other parts of the body effectively.

Different penetration profiles may result in parts of the body that consistently have drug concentrations that allow the replication of partially resistant genotypes (*Figure 4C*). A partially resistant virus migrating or reactivating from the latent reservoir in that part of the body may establish a growing HIV population. Even if these variants can only successfully replicate in one part of the body (i.e. the single drug compartment), they may still lead to a detectable viral load in a blood sample. After establishing in a single drug compartment, variants have the opportunity to acquire additional mutations that allow them to replicate elsewhere. In principle, this can explain the three observations above. First, drug resistance can continue to evolve at rates determined by the population mutation rate and migration rates into such a single-drug compartment even after years of virologic suppression. Second, partially resistant viruses may be observed if replication in one or more single drug

compartments can be detected in the blood. Third, drugs with different penetrance profiles can explain why drug resistance evolves in a predictable order - namely in response to the best-penetrating drug first.

As in the temporal heterogeneity model, the spatial heterogeneity model may help us understand qualitative patterns previously observed in clinical trial data. In the Kempf study, patients were treated with 3TC, NFV, and D4T (see *Figure 1A*), where a plausible 3TC single-drug compartment may exist. 3TC, NFV, and D4T have differential rates of penetrance into the brain, with penetrances of NFV and D4T particularly poor (*Letendre et al., 2008*; *Browne et al., 1993*; *Aweeka et al., 1999*; *Ene et al., 2011*). This can create parts of the brain where 3TC-resistant HIV can effectively replicate and spread, potentially allowing detection in the blood as a 3TC-resistant type. Once resistant to 3TC, the virus needs only one additional mutation to replicate in the presence of NFV. In practice, because of D4T's poor potency (*Rosenbloom et al., 2012*), much of the body may effectively be a 3TC+NFV compartment and select for NFV mutations next (*Figure 1A*).

To test whether a model of spatial heterogeneity of drug levels can explain the three clinically observed features, we extended a pre-existing model of HIV spatial drug penetrance to account for three drug combinations (*Moreno-Gamez et al., 2015*). Similar to the temporal model, we simulated 1500 individuals infected with HIV and treated with 3TC+D4T+NFV and followed for up to 10 years. Here, instead of drugs decaying over time, we computed the average drug dose over time in a fully adherent patient, but assumed that drugs do not reach all parts of the body equally. We simulated three compartments interconnected by migration: (1) the majority of the body reached by all three drugs at full concentration, (2) a brain-like compartment reached by all three drugs at lower concentrations according to comparative measurements of drug levels in the plasma and cerebrospinal fluid (CSF) (*Ene et al., 2011*) and (3) a small sanctuary compartment reached by no drugs. We recorded when the viral population in the main compartment (presumably the measured compartment when assaying viral loads) exceeded a detectable threshold (see Materials and methods).

We found that this model of spatial heterogeneity produced all three features that we observed in the clinical data under three-drug combination therapies, but only for one of the drugs in the combination - 3TC (*Figure 5*). First, 3TC resistance continues to emerge, even if individuals have been on therapy for several years. This decline in probability over time emerges from the simulated distribution of viral setpoints, which vary between $10^4$ and $10^6$ (interquartile range $7.92 \times 10^4 - 3.18 \times 10^5$). Individuals with larger population sizes evolve resistance more rapidly than those with smaller population sizes (*Figure 5—figure supplement 1*), and simulating a narrower band of viral set points (interquartile range $1.36 \times 10^5 - 1.71 \times 10^5$) results in a yearly resistance probability that remains approximately constant over time (*Figure 5—figure supplement 2*). Second, resistance mutations exclusively emerge one at a time and can lead to clinically detectable levels. Third, resistance mutations occur in a predictable order, and we see overwhelmingly resistance to 3TC and no ongoing resistance to NFV and D4T. We see only 3TC resistance evolve because the penetrance of D4T and NFV in the CSF is sufficiently low that the cost of D4T or NFV resistance dominates over its benefit (i.e. $R_{NFV/D4Tresistant} < R_{NFV/D4Tsensitive}$). This suggests that compartments with additional patterns of drug penetrance are necessary to provoke additional resistance evolution. Even though this simplified model of drug spatial heterogeneity does not capture additional resistance evolution for two drugs in the combination, all three patterns straightforwardly emerge with respect to 3TC resistance (*Figure 5*).

Above, we examined a PI-based therapy, but a similar spatial model may also explain some of the patterns in NNRTI-based therapies. NNRTIs tend to have strong penetration profiles, which may explain why drug resistance to NNRTIs emerges first (*Figures 1B* and *2C*). However, many NNRTIs are highly suppressive even at relatively low concentrations and sometimes even for genotypes with resistance mutations. This suggests that an NNRTI resistance evolution may require not only a single drug compartment (i.e. the absence of other drugs), but also a sub-clinical concentration of the NNRTI to select for resistance mutations.

For both the spatial and the temporal models there is an interesting and somewhat counter-intuitive effect. Intuitively, one might think that resistance will first arise in response to the 'worst' drug in a cocktail (least potent, shortest half-life, or worst penetration profile). However, in a model of spatial or temporal heterogeneity, a desirable property –high penetration or long half-life– actually makes a drug vulnerable to resistance. Therefore, the best way to prevent the evolution of 3TC

resistance (when treatment is with 3TC+NFV+D4T) is not to target the shortcomings of 3TC, but to target the shortcomings of the other drugs. Specifically, if 3TC monotherapy occurs in space or time because of a long half-life or good penetration, the solution would be to add another drug that also has a long half-life or good penetration.

The introduction of second generation PIs in the early 2000s resulted in exactly this intervention. In particular, ritonavir-'boosted' lopinavir (LPV/r) (*Tseng et al., 2015*) resulted in improved penetance and extended half-life in comparison to NFV (*Capparelli et al., 2005*). The clinical effect of replacing NFV with LPV/r in the previously described Kempf study (*Kempf et al., 2004*) is a substantial reduction of resistance evolution by 76%. In the NFV arm of the study, 79 of 327 individuals developed some type of drug resistance (24%) within 2 years, whereas in the LPV/r arm, only 19 of 326 did (6%).

In the context of resistance evolution via spatially-varying drugs, LPV/r consistently reaches a level above the 50% inhibitory concentration in cerebrospinal fluid (*Capparelli et al., 2005*), thereby likely shrinking the size or number of compartments in which 3TC exists alone. The fourfold lower rate of drug resistance evolution among individuals on the LPV/r regimen suggests that the enhanced spatial coverage of LPV/r decreased the portions of the body where 3TC reached alone by a factor of four. In the context of resistance evolution via temporally varying drugs, the longer half-life of LPV/r could reduce the time of 3TC as effectively a monotherapy. Future models should test whether differences in drug penetrances or half-lives better explain the reduced rate of resistance with LPV/r treatments.

## Discussion

Above we have described ideas about how drug-level heterogeneity in time, due to different half-lives and non-adherence, can lead to periods of monotherapy, which, in principle, can lead to drug resistance evolution during triple-drug therapy. We have also described how heterogeneity in space, due to different penetration profiles of drugs, can lead to single-drug compartments, which also, in principle, can lead to drug resistance evolution. We extended existing one or two drug models for imperfect adherence and imperfect drug penetrance and found that a model of spatial heterogeneity can recreate the observed clinical patterns – ongoing, sequential and predictable evolution – for resistance to specific drugs but not others. A temporal model readily matches certain observations (sequential and predictably ordered mutations), but not others (namely, ongoing resistance after years on therapy). Therefore, we have two possible explanations for the observation that drug resistance can evolve during triple-drug therapies, neither of which currently explain all aspects of the data in their current state. Further, while we know that temporal and spatial heterogeneity exist and *can* cause drug resistance evolution, this does not prove that they do.

Models that are intuitively convincing need not be true. For example, NNRTIs have long half-lives, which can lead to periods of monotherapy (the so-called 'NNRTI tail') if patients miss doses or interrupt treatment. Additionally, HIV often evolves resistance to NNRTIs (*Rocheleau et al., 2018*). Yet, there is some evidence that the NNRTI tail does not explain NNRTI drug resistance in low-adherence patients (see discussion under temporal heterogeneity). In our model of temporal heterogeneity in drug levels through incomplete adherence (of PIs, not NNRTIs), many individuals stayed virologically suppressed despite poor adherence. Even though drug conditions intermittently select for drug resistant variants, viral populations could not reach critical population sizes rapidly enough to produce the relevant variants that could spread. While outside the scope of this study, we propose that a quantitative approach that includes mutant selection windows, pharmacokinetics and population genetics is needed to adjudicate between potential explanations for a variety of drug combinations. Can we build a model that can quantitatively match the observations?

Detailed models of HIV drug resistance have already been developed, especially for models of temporal heterogeneity. For example, in a carefully parameterized HIV model focused largely on single drugs (*Rosenbloom et al., 2012*), the authors examined how adherence patterns for a two-drug therapy (boosted darunivir and raltegravir, DRV/r + RAL) affected the mode and rate of resistance evolution. They found that virologic failure could occur from a viral population resistant to raltegravir but not darunivir, because of the characteristics of the darunivir mutant selection window. Similarly, *Bershteyn and Eckhoff, 2013* examined drug resistance dynamics in individuals taking PrEP (TDF) under a realistic model of non-adherence. Adherence models that more closely mimic human

behavior may allow for ongoing evolution of resistance years after initial virologic suppression, a clinical outcome expected but not observed in our exploration of 3TC+D4T+NFV. However, these analyses of one or two drug combinations served as proofs of concept, and no three-drug therapies were examined. A real gap therefore remains in developing realistic models that can explain complex patterns of resistance emergence under the most widely used triple-drug therapies.

For the spatial models, the gap between the models and clinical data is even larger than for the temporal models, because the models that have been studied are simpler (*Perelson et al., 1997*; *Moreno-Gamez et al., 2015*; *Lorenzo-Redondo et al., 2016*). Filling this gap will be particularly challenging because many parameters remain unknown. Understanding where, when, and how certain drugs reach different parts of the body is complicated (*Vendel et al., 2019*) and indeed, different studies have found different relative penetrances of drugs into the tissue versus the plasma. In addition, studies of drug penetrance may not be quantifying the correct factors to help us understand where drug resistance can emerge. First, the administered version of a drug may be at a different concentration than the active form (*Dumond et al., 2008*). Second, the intracellular concentration of a drug, rather than the tissue concentration, may mediate effectiveness, requiring that we understand not only whether drugs reach a tissue, but the rates at which they are transported into cells (*Bazzoli et al., 2010*; *Dumond et al., 2008*). Third, we may not be investigating penetrance in the correct compartments, or at the correct spatial scale. For example, there can be considerable heterogeneity of drug penetrance even within a single lymph node or gut cross-section (*Thompson et al., 2015*; *Thompson et al., 2019*). Fourth, many studies report tissue drug concentrations relative to the blood plasma, without considering the actual tissue drug level and its inhibitory effect on the virus (*Else et al., 2011*). Even comparing tissue drug concentration to the IC50 does not capture the full dynamics of drug concentrations within the body (*Shen et al., 2008*).

For both the spatial and temporal models, differences between people should also be taken into account. For example, treatment with other drugs (e.g. tuberculosis drugs) may affect the half-lives and penetration of HIV drugs (*López-Cortés et al., 2002*). Genetic variation among treated individuals may also mediate penetrance (see *Apostolova et al., 2015*). For example, efavirenz concentrations are higher in people with certain genotypes, and these genotypes are more common in Black/African groups. Rates of drug metabolism also differ between men and women (*Else et al., 2011*). Different adherence patterns could hypothetically make different mutational orderings more or less likely as well. A recent paper suggests that the combination of imperfect adherence with imperfect penetration makes sequential drug resistance evolution likely in an in vitro system (*Lustig et al., 2019*).

For now, it remains unclear if temporal heterogeneity, spatial heterogeneity, or a combination of the two cause the evolution of drug resistance during triple-drug therapy. Clarity on this important unresolved mystery will aid the rational design of treatments that minimize the risk of resistance evolution.

Mathematical modeling has played an essential role throughout the HIV pandemic, and we do not intend to diminish these accomplishments. Early studies set a gold standard in uniting available measurements of viral populations (most often viral load) with quantitative models to understand many facets of HIV biology (see review in *Nowak and May, 2000*). Indeed, these studies helped motivate the need for triple-drug therapies in the first place (*Perelson and Nelson, 1999*). Now, armed with decades of genetic data, we need to continue this work to understand why triple-drug therapy is not evolution-proof.

Future studies addressing the evolution of multi-drug resistance in HIV should use realistic parameters (for example, like *Rosenbloom et al., 2012* and *Bershteyn and Eckhoff, 2013*) and will need to grapple with the pharmacokinetics of multiple drugs and how they interact (*Koizumi and Iwami, 2014*; *Jilek et al., 2012*). Experimental work has highlighted drug pairs that suppress in vitro viral populations to both greater and lesser extents than each drug working independently, which may provide a partial answer to why some combinations of drugs fail at a surprisingly high rate (*Jilek et al., 2012*). These future studies should not just match viral loads and rates of patient failure, but also genetic patterns of resistance. In addition to the patterns described in this paper (ongoing, sequential, and predictable resistance evolution), other observations should be considered: drug resistance rates have fallen over time (*Feder et al., 2016a*), virologic failure varies as a function of adherence (*Harrigan et al., 2005*) and between different drug combinations (*Lee et al., 2014*; *Feder et al., 2016a*). We hope that expanded models will allow us not only to make predictions –

for example, about rates and patterns of resistance for novel combinations of drugs – but also to learn more about HIV's biology from data we already have. For example, can we use patterns of resistance to estimate how large single-drug compartments are, or how rapidly viruses migrate between different parts of the body (*Feder et al., 2019*)?

Drug resistance represents a serious threat across disease systems, and spatial and temporal variation in drug levels likely contribute in these other instances as well. For example, while many in the cancer community argue that applying multiple drugs simultaneously should halt the evolution of resistance (*Basanta et al., 2012*), experimental evidence also suggests multi-drug resistant cells may already exist as standing genetic variation (*Bhang et al., 2015*). In treated solid tumors, drugs that cannot penetrate through the entire tumor may form a concentration gradient of selection pressure that theoretical (*Hermsen et al., 2012*; *Fu et al., 2015*; *Gralka et al., 2017*) and experimental (*Baym et al., 2016*) evidence suggests may speed the evolution of resistance. In tuberculosis, there is evidence that drug levels in various types of granulomas are often not as high as in the blood (and do not reach bactericidal levels) (*Prideaux et al., 2015*), and may be particularly low for some drugs in cavitary lesions (*Strydom et al., 2019*). Since not all tuberculosis patients have cavitary lesions, this means that ideal drug choice or dosing may not be the same for all individuals (*Strydom et al., 2019*). Lessons about spatial and temporal heterogeneity in drug concentrations in HIV can form a scaffold upon which we can try to understand resistance evolution in other disease systems.

Even apart from disease, every natural system experiences multiple stressors simultaneously, even if those stressors are less extreme than targeted anti-retrovirals. How populations evolve in response to complex environments represents a fundamental question in evolutionary biology. In addition to (and perhaps because of) its medical relevance, HIV represents a powerful study system for interrogating this question. We understand many of its population parameters relating to viral kinetics, and extensive research has produced a well-developed genotype-phenotype map of resistance mutations. Because drug resistance emerges repeatedly in different individuals, we can understand which features of this intra-patient evolution are generalizable, which provides insights into how evolution mediated by temporally and spatially varying selection pressures occurs.

In theory, multi-drug therapies should suppress pathogen populations and prevent the evolution of resistance. In practice, multi-drug therapies are challenging to achieve, and pathogen populations do not experience all drugs in all places at all times. When diagnosing the failure of any multi-drug approach because of resistance evolution (whether in cancer, antibiotics, herbicides, etc), we suggest the following four guiding questions: (1) Are all drugs effective? (2) Is there cross-resistance? (3) Is there spatial heterogeneity in drug levels? (4) Is there temporal heterogeneity in drug levels? Through understanding the ways that the application of multi-drug therapies falls short of the theory, we are poised to address those shortcomings and decelerate the evolution of resistance.

## Materials and methods

### Measuring rates of drug resistance evolution over time (*Figure 2A*)

To understand how an individual's probability of acquiring resistance changes over time, we reanalyzed data from a Canadian cohort study (*Rocheleau et al., 2018*). This is a cohort of individuals with HIV taking combination ART for the first time. We would like to understand the probability that a given individual acquires a drug resistance mutation in a given year, conditional on having acquired no resistance mutations to any class of drugs prior to that year. To estimate this probability, we performed a conditional survival analysis using the R package condSURV (*Meira-Machado and Sestelo, 2016*). We computed separate estimates for each year following the start of therapy and for each drug resistance mutation class (3TC, NRTI, NNRTI, or PI) for the probability of acquiring a resistance mutation conditional on having acquired no resistance mutations prior to that year. Within each analysis, individuals acquiring a drug resistance mutation for a specific class of drugs were coded as endpoint events, and individuals leaving the study for all other reasons (including resistance to other drugs) were censored. To ensure adequate follow-up time, we analyzed only individuals beginning therapy between 1996 and 2005 ($N = 2692$).

In *Figure 2*, we fit exponential functions to the conditional survival point estimates for each drug resistance class (3TC, other NRTI, NNRTI, PI) using the formula $f(t) = \alpha e^{\beta t}$. Parameter fits for $\alpha$ and $\beta$ are reported in *Supplementary file 3*.

## Determining the total number of resistance mutations and the first drug resistance mutation on a treatment (*Figure 2B and C*)

To determine the number of mutations at sampling and the identity of the first mutation (if present) for a variety of triple-drug therapies, we analyzed data from *Feder et al., 2016a* representing Sanger-sequenced HIV populations treated with a broad range of regimens between 1989 and 2013 (Stanford HIV Drug Resistance Database). We included only individuals treated with exactly one regimen of three drugs to minimize pre-existing resistance before therapy onset. Ambiguous underlying nucleotide calls (i.e., non-A/T/C/G calls) were interpreted as population polymorphisms among all possible resulting amino acids. For example, an AAS residue (AAC/AAG) was recorded as an asparagine/lysine polymorphism, but an AAY (AAC/AAT) was recorded as asparagine. When multiple sequences were available for the same individual at the same time point, polymorphisms were also recorded.

We then determined the number of drug resistance mutations by comparing sequences to the WHO list of surveillance drug resistance mutations (DRMs) (*Shafer et al., 2007*). DRMs were only counted if HIV sequences contained residues conferring resistance to the therapy with which they were treated. Two classes of DRMs were recorded: (1) polymorphic DRMs in which some calls supported the DRM and some supported a non-DRM and (2) non-polymorphic DRMs, in which all calls supported the DRM.

To determine the first DRM, we retained only individuals with exactly one polymorphic DRM or exactly one non-polymorphic DRM and any number of polymorphic DRMs (under the assumption that fixed mutations occurred before polymorphic mutations). For individuals with sequences taken at multiple time points, we retained the first sample meeting the conditions above (one polymorphic DRM or one fixed DRM and any number of polymorphic DRMs). In *Figure 2B*, we plot all three drug treatments with at least 25 individuals. In *Figure 2C*, we plot all three drug treatments with at least 10 individuals with a single drug resistance mutation.

## Testing the enrichment of drug resistance mutation classes in combination therapies (*Figure 2C*)

To test the hypothesis that mutations conferring NNRTI resistance occur first on NNRTI-3TC/FTC-NRTI therapies and M184V/I (conferring resistance to 3TC/FTC) occurs first on PI-FTC/3TC-NRTI-based therapies, we first calculated a baseline expectation for the relative rates of resistance emergence for each drug in a three drug combination therapy based on mutation rates alone. We based our calculations on the consensus subtype B sequence provided by the Los Alamos HIV Database (https://www.hiv.lanl.gov/content/index), the WHO list of resistance mutations (*Shafer et al., 2007*), and mutation rates estimated in *Abram et al., 2010*. For each relevant amino acid position in protease and reverse transcriptase proteins, we determined all possible one-step mutations. For all one-step mutations, we determined whether they led to resistance to any drugs according to the WHO list. Then, for each drug, we summed the mutation probability for all possible one-step mutations that lead to resistance to that certain drug. The resulting total mutation probabilities ranged from $4 \cdot 10^{-5}$ for TDF to $2 \cdot 10^{-4}$ for LPV. The calculated values are listed in *Supplementary file 4*.

We then used the drug-specific mutation rates to calculate the expected frequency of resistance to one class of drugs versus the others. Specifically, for NNRTI-based therapies, we divided the sum of the mutation rates for the NNRTI-resistance conferring mutations by the total mutation rate for all of the drug resistance mutations in a given therapy. For PI-based therapies, we divided the mutation rate for 3TC or FTC resistance-conferring mutations by the total mutation rate for all of the drug resistance mutations in a given therapy. We used a one-sided binomial test to determine if the number of individuals with NNRTI resistance or 3TC/FTC resistance as their first mutation was higher than expected by chance on therapies composed of NNRTI-3TC/FTC-NRTI and PI-FTC/3TC-NRTI, respectively. We used a Benjamini-Hochberg correction with a 5% false discovery rate separately for the NNRTI- and PI-based hypotheses. Full details on the tests performed, including the mutation-rate-based null hypotheses for each therapy and the number of successes and failures are listed in *Supplementary files 5* and *6*.

## Modeling temporal heterogeneity (*Figure 5A*)

We extend the viral dynamics model in *Rosenbloom et al., 2012* to consider the application of three drugs simultaneously. In this model, viral particles have a reproductive number $R$ that governs their rate of replication. The presence of drugs alters $R$ as a function of drug concentration, and drug resistance changes the shape of this curve (*Figure 3B*). Generally, these resistance mutations carry a cost that reduce $R$, although in the presence of drugs, these drug resistance mutations confer a fitness advantage.

### Effect of a drug on viral reproductive number

Following *Rosenbloom et al., 2012*, the effect of a drug on viral fitness is computed as a function of the concentration $[x]$ of drug $D$ if the virus does not possess any resistance mutations:

$$r_{sensitive,D}([x]) = \frac{1}{1 + (\frac{[x]}{IC_{50,D}})^{m_D}}$$

(1)

and

$$r_{resistant,D}([x]) = \frac{(1 - s_D)}{1 + (\frac{[x]}{\rho_D IC_{50,D}})^{m_D(1+\sigma_D)}}$$

(2)

if the virus possesses a mutation conferring resistance to drug $D$. The parameters $\rho_D, IC_{50,D}, m_D, s_D, h_D$ and $C_{max,D}$ (see below for $C_{max,D}$ and $h_D$) are specific to particular drug/mutation combinations (*Rosenbloom et al., 2012*), and are given in *Supplementary file 7*. Since multiple drug resistance mutations confer resistance to a given drug, we created a single composite drug resistance mutation with the most favorable attributes for drug resistance evolution of all possible single-step resistance mutations - that is, highest mutation rate μ, lowest cost of resistance $s$, highest ρ and lowest σ (following *Rosenbloom et al., 2012*). For NFV, this resistance mutation ends up equivalent to D30N and for D4T, this resistance mutation ends up equivalent to M41L. For 3TC, this resistance mutation is equivalent to M184V with a slightly lower cost of resistance (0.41 versus 0.46) taken from K65R. Therefore, the basic reproductive ratio of a virus of genotype $i$ in the presence of drug $D$ at concentration $[x]$ is as follows:

$$r_{i,D}([x]) = \begin{cases} r_{sensitive,D}([x]) & \text{if i is sensitive to D} \\ r_{resistant,D}([x]) & \text{if i is resistant to D} \end{cases} .$$

To determine the relevant drug concentration some number of hours after dosing, we assume that the drug decays exponentially from a maximum concentration $C_{max,D}$ at 0 hours after dosing with half-life $h_D$:

$$c_D(t) = C_{max,D} 2^{-t/h_D}.$$

(3)

This is evaluated in half hour increments to compute the concentration of each drug over time.

To simulate lack of adherence, at each dosing period, an individual does not take the required drugs with probability $p$. Failure to take a dose is evaluated separately for each drug, following *Rosenbloom et al., 2012*. Additionally, to simulate longer blocks of non-adherence related to failures to fill prescriptions, each year, individuals also have a $p$ probability of an additional two week treatment interruption with no doses taken (following ideas discussed in *Bershteyn and Eckhoff, 2013*). We model the adherence probability $p$ to be uniformly distributed between 0 and 1, although empirical measurements suggest the distribution should be weighted more heavily toward higher adherences (*Bangsberg et al., 2006*).

We compute the basic reproductive ratio of genotype $i$ at time $t$ under a combination of multiple drugs as the product of the reproductive numbers against individual treatments times a scaling factor $R_{00}$ ($R_{00} = 10$, following *Rosenbloom et al., 2012*):

$$R(i,t) = R_{00} \prod_{D \in (3TC,D4T,NFV)} r_{i,D}(C_D(t)).$$

(4)

This formulation assumes that the drugs affect viral fitness Bliss-independently. *Jilek et al., 2012*

has measured that 3TC and D4T combine to be more potent than Bliss-independence, but NFV/3TC and NFV/D4T combinations are less potent than Bliss-independence (although more potent than Loewe additivity), suggesting that Bliss-independence is a reasonable assumption for the three drug combination therapy. *Equation 4* allows us to compute the basic reproductive number for all eight genotypes (conferring resistance to all combinations of 0, 1, 2, or 3 drugs) at all points in time.

## Stochastic simulation algorithm

### Model setup

First, we determine when a completely adherent individual would take all doses for a 10 year period. Note, 3TC and D4T are dosed twice a day, and NFV is dosed three times a day. We then binomially sample which doses the individual misses based on their adherence probability $p$, and compute the corresponding drug concentrations for all three drugs at all half hour intervals following *Equation 3*. From these drug concentrations, we compute the basic reproductive ratio for each of the eight genotypes at each half hour interval following *Equation 4*.

We next determine the frequency of drug resistance mutations as standing genetic variation in the viral reservoir established before the application of drugs. The frequency of each mutation is determined by the ratio between the rate at which it is produced via mutation ($\mu_D$) and its cost in the absence of the drug ($s_D$), or 'mutation-selection balance' ($f_D = \mu_D/s_D$). The frequency of the drug-sensitive allele for a given drug $D$ is $1 - f_D$. We determine the frequency of the eight combinations of drug resistance mutations in the reservoir, $f$, by computing the product of the three drug resistant or sensitive allele frequencies at mutation-selection balance for a given genotype $i$. Note, stochastic effects result in mutational frequencies that can vary substantially around $\mu/s$ (*Hermisson and Pennings, 2017*), but this stochasticity is not incorporated here. The initial population size of each of the eight genotypes is set to be $y(0) = f * N_e$, rounded to the nearest integer, where $N_e$ is drawn from a lognormal distribution. To generate values of $N_e$, we draw an exponent from a normal distribution with mean 5.17 and standard deviation 0.5, and exponentiate 10 by this value. This distribution is centered on the estimated short-term viral effective population size, $1.5 \times 10^5$ (*Pennings et al., 2014*), but varies to reflect individual variation (first decile: $7.5 * 10^4$, ninth decile: $3.3 \times 10^5$). We redraw any population sizes less than $10^4$ or greater than $10^6$.

### Iterated step

At each timestep of length 30 min ($\Delta t = 0.02$), a Poisson-distributed number of cells of each genotype, $e_i$, emerges from the latent reservoir with mean proportional to the genotype's frequency in the reservoir $f_i$:

$$e_i \sim Poisson(A\Delta t f_i). \tag{5}$$

$A$ represents the number of cells expected to reactivate from the latent reservoir in a day, and we set $A = 3000$ following *Rosenbloom et al., 2012*.

Each of eight genotypes $i$ produces a Poisson distributed number of newly infected cells ($n_i$) with mean

$$n_i \sim \Delta t \left( \frac{\lambda R(i,t) d_y y_i}{\lambda + \sum_{j=1}^{n} R(j,t) d_y y_j} \right) \tag{6}$$

where $R(i,t)$ is computed as in *Equation 4*, $d_y$ represents the death rate ($d_y = 1$, representing a average lifespan of 1 day), and $\lambda = N_e * R_{00}/(R_{00} - 1)$ (*Rosenbloom et al., 2012*). Each of these newly infected cells has a probability of mutating to acquire or lose drug resistance to drug $D$ according to mutation rate $\mu_D$ independently at each of its three loci. We use $mut_{i \to j}$ to denote the number of cells mutating from genotype $i$ to genotype $j$ at the point of infection.

Before cell death, the number of actively infected cells of genotype $i$ is given as follows:

$$a_i = y_i(t) + e_i + n_i + \sum_j mut_{j \to i} - \sum_j mut_{i\% \to j}. \tag{7}$$

Each cell has probability of dying $p = (1 - e^{-dy\Delta t})$, resulting in $d_i \sim binom(p, a_i)$ deaths of cells of

genotype $i$. Therefore, the number of cells of genotype $i$ after one simulation step is given as follows:

$$y_i(t + \delta t) = a_i - d_i. \tag{8}$$

## End condition

Patients are evaluated every 3 months (12 weeks) for 10 years (480 weeks). If the patient had more than 5000 actively infected cells at an evaluation time point (indicating a substantial increase above baseline reservoir reactivation), the viral population was assessed to have undergone virologic rebound, and the simulation is halted. At each of the three drug resistance loci, drug resistance mutations were recorded if they comprised at least 15% of the population.

## Modeling spatial heterogeneity (*Figure 5B*)

The spatial model analyzed extends *Moreno-Gamez et al., 2015* to consider the application of three simultaneous drugs. We also used more realistic fitness values (described above) which reflect that even drug resistant viruses have reduced fitness in the presence of therapy. In this model, we simulate three compartments with different degrees of drug penetrance: (1) a main compartment where all three drugs are present with a carrying capacity $K$ drawn from a log-normal distribution as described above, (2) a brain-like compartment with reduced drug concentration according to experimental measurements (*Ene et al., 2011*, *Supplementary file 7*) [because cerebrospinal fluid represents approximately 3% of the plasma volume, this second compartment is 3% of the size of the first compartment], and (3) a sanctuary compartment where no drugs reach, which is 1% of $K$. Note, the true sizes of these compartments are unknown. In *Figure 5—figure supplement 2*, we also simulate $K$ drawn from a narrower distribution with an exponent (base 10) drawn from a normal distribution with mean 5.17 and standard deviation 0.05 (an order of magnitude smaller than the standard deviation of 0.5 simulated in *Figure 5B*).

Unlike the temporal model, we assume for the spatial model that drug levels are constant throughout therapy, and are set to be the average drug levels across a single dosing period in an individual with perfect adherence (see *Supplementary file 7* for reference values). We compute the average drug level over a full dosing period ($t_D = 12$ hr for 3TC and D4T, 8 hr for NFV):

$$\epsilon_D = 1 - t_D^{-1} \int_0^{t_D} c_D(t) dt. \tag{9}$$

## Effect of a drug on viral reproductive number

Because the virus can be resistant or sensitive to each of three drugs, there are eight relevant genotypes. In each of the three drug compartments, we compute the effective reproductive number of all eight genotypes given the levels of drugs present. For a virus with genotype $i$ in a compartment $j \in$ (Sanctuary, CSF, Plasma), $R$ is computed as follows:

$$
\begin{aligned}
R(i, \text{Sanctuary}) &= R_{00} \prod_{3TC, D4T, NFV} r_{i,D}(0) \\
R(i, \text{CSF}) &= R_{00} \prod_{3TC, D4T, NFV} r_{i,D}(p_D[\epsilon_D]) \\
R(i, \text{Plasma}) &= R_{00} \prod_{3TC, D4T, NFV} r_{i,D}([\epsilon_D])
\end{aligned} \tag{10}
$$

where $r_{i,j}([x])$ represents the ability of genotype $i$ to replicate in the presence of drug $j$ at concentration $[x]$ (*Equation 5*), $\epsilon_D$ represents the drug level in a compartment (*Equation 9*) and $p_D$ represents the average proportion of plasma drug concentrations for drug $D$ measured in the CSF (*Ene et al., 2011*). As in the temporal model, $R_{00} = 10$. Drug-specific model parameters are given in *Supplementary file 7*.

## Stochastic simulation algorithm

## Model setup

We instantiate the simulations with the sanctuary compartment at carrying capacity, and the other two compartments empty. No resistance mutations are present in the initial population.

## Iterated step

The rate of an event is $\alpha_{i,j,k}$, where $i = \{0, 1, 2, 3, 4, 5, 6, 7\}$ is the genotype of the cell undergoing the event (wildtype, 3TC resistant, D4T resistant, NFV resistant, 3TC+D4T resistant, 3TC+NFV resistant, D4T+NFV resistant, 3TC+D4T+NFV resistant), $j = \{0, 1, 2\}$ corresponds to the compartment in which the event will occur (plasma, CSF, sanctuary) and $k = \{0, 1, 2\}$ corresponds to the type of event - replication, death or migration. $\alpha_{i,j,k} = N_{i,j} * \lambda_{i,j,k}$, where $N_{i,j}$ is the number of cells of a type in a given compartment and $\lambda_{i,j,k}$ is the rate at which event $k$ occurs for cells of type $i$ in compartment $j$.

- If $k = 0$, the virus infects a new cell, which happens with rate $\lambda_{i,j,0}$ computed as in **equation 10**. Three random uniform numbers $X_A, X_B, X_C \in [0, 1]$ are drawn, corresponding to potential mutations to drugs $A$, $B$ and $C$. For each drug $D$, if $X_D < \mu_D$, the virus mutates from genotype $i$ to the alternative resistance state for the given drug $i'$. This allows both back mutation, and multiple resistance mutations to emerge in a single round of replication. $N_{i,j}$ is decremented and $N_{i',j}$ is incremented.
- If $k = 1$, the cell dies, which happens with rate $\lambda_{i,j,1} = 1$, and $N_{i,j}$ decreases by 1.
- If $k = 2$ the cell migrates from compartment $j$ to an alternate compartment $j'$ with equal probability, which happens with rate $\lambda_{i,j,2} = 0.1$ (approximately the estimated plasma to lymph migration rate in *Feder et al., 2019*). In the case of migration, $N_{i,j}$ decreases by 1 and $N_{i,j'}$ increases by 1. Once a cell has been selected to migrate, it migrates to an alternate compartment with equal probability.

If the evaluation of an event causes a compartmental population size to grow larger than the compartment-specific carrying capacity, cells in that compartment are downsampled multinomially so that the carrying capacity is enforced.

## End condition

Patients are evaluated every 3 months (12 weeks) for 10 years (480 weeks). If the patient has more than 150 actively infected cells at an evaluation time point, the viral population was assessed to have undergone virologic rebound, and the simulation is halted. The reasoning for this end condition is as follows: the viral setpoint is approximately $10^5$ copies/mL and is readily detectable in the blood. 100 copies/mL is an approximate cutoff for detectable virus in the blood, and is 0.1% of the viral setpoint. We therefore used a cutoff 0.1% of the average simulated effective population size as a detectability threshold. At each of the three drug resistance loci, drug resistance mutations were recorded if they comprised at least 15% of the population.

### Code availability

Code to run all data analysis and simulations is available at https://github.com/federlab/HIV-MDR-evolution/ (*Feder et al., 2021b*). Data to reproduce Figure 3B and C is from *Feder et al., 2016a* and is also available on GitHub (*Feder et al., 2016b*).

## Acknowledgements

We thank Alison Hill, Richard Harrigan, Jonathan Pritchard, Noah Rosenberg, Dmitri Petrov, Nandita Garud, Christopher McFarland, Ambika Kamath, Stephen Martis, and three reviewers for valuable feedback on earlier versions of this manuscript. We also thank Noah Rosenberg and Jonathan Pritchard for encouraging us to write this paper. We thank Chris Hoffmann for providing help interpreting the raw data from his 2009 CID paper and Philana Ling Lin for consulting about the tuberculosis discussion. AFF is supported by a fellowship from the Miller Institute for Basic Research in Science. PSP is supported by the National Science Foundation, grant number 1655212 and the National Institute of Health, grant number R01AI134195.

## Additional information

### Competing interests

Kristin N Harper: is an employee of Harper Health & Science Communications, LLC. The author declares that no other competing interests exist. The other authors declare that no other competing interests exist.

### Funding

| Funder | Grant reference number | Author |
|---|---|---|
| National Science Foundation | 1655212 | Pleuni S Pennings |
| National Institutes of Health | R01AI134195 | Pleuni S Pennings |
| Miller Institute for Basic Research in Science | | Alison F Feder |

The funders had no role in study design, data collection and interpretation, or the decision to submit the work for publication.

### Author contributions

Alison F Feder, Pleuni S Pennings, Conceptualization, Formal analysis, Visualization, Writing - original draft, Writing - review and editing; Kristin N Harper, Writing - original draft, Writing - review and editing; Chanson J Brumme, Formal analysis

### Author ORCIDs

Alison F Feder (ID) https://orcid.org/0000-0003-2915-089X
Pleuni S Pennings (ID) https://orcid.org/0000-0001-8704-6578

### Decision letter and Author response

Decision letter https://doi.org/10.7554/eLife.69032.sa1
Author response https://doi.org/10.7554/eLife.69032.sa2

## Additional files

### Supplementary files

• Supplementary file 1. We briefly review salient existing models that investigate the evolution of HIV resistance evolution via drug heterogeneity in time or space.

• Supplementary file 2. Conditional survival probabilities for resistance evolution in a large cohort of HIV-infected individuals in British Columbia treated with three-drug combination therapies.

• Supplementary file 3. Exponential fits for conditional survival probabilities in *Figure 2A*. Variables in table are fit as follows $f(t) = \alpha e^{\beta * t}$.

• Supplementary file 4. Computed rates toward single drug resistance for anti-retrovirals assessed in *Figure 2C*. See materials and methods section 'Determining the total number of resistance mutations and the first drug resistance mutation on a treatment' for more details.

• Supplementary file 5. Binomial test summaries to assess if 3TC resistance precedes other categories of resistance in individuals treated with PI-based combination therapies.

• Supplementary file 6. Binomial test summaries to assess if NNRTI resistance precedes other categories of resistance in individuals treated with NNRTI-based combination therapies.

• Supplementary file 7. Drug-specific model parameters for 3TC, NFV and D4T.

• Transparent reporting form

## Data availability

Data to reproduce Figure 3A is protected patient information, but code to run the analysis is available at the GitHub URL below and the script output (which is plotted in Figure 2A) is available in Supplementary file 2. Data to reproduce Figure 3B and C is from Feder et al, 2016, and is also available on GitHub: https://github.com/affeder/HIV-DRM-sweeps/blob/master/dat/dataset.01.24.txt. Code to reproduce all analyses is available on GitHub: https://github.com/federlab/HIV-MDR-evolution (copy archived at https://archive.softwareheritage.org/swh:1:rev:ae825d43929d9334d6b0aa41695956cf4492dccf).

The following datasets were generated:

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
