## [Decision Letter]

**Acceptance summary:**

This study considers how HIV evolutionary dynamics in a multiple drug-treated individual can give rise to the clinical patterns of the accrual of drug resistance mutations, including with understandings of the pharmacokinetics of the drugs in the body to help explain some of the patterns. The subject is of importance both clinically – for the optimal treatment choice for people living with HIV – and scientifically, due to the potential to predict and interpret evolutionary trajectories.

**Decision letter after peer review:**

Thank you for submitting your article "Understanding patterns of HIV multi-drug resistance through models of temporal and spatial drug heterogeneity" for consideration by *eLife*. Your article has been reviewed by 3 peer reviewers, and the evaluation has been overseen by George Perry as the Senior and Reviewing Editor. The following individuals involved in review of your submission have agreed to reveal their identity: Danna Gifford (Reviewer #1); Katherine Atkins (Reviewer #3).

Essential revisions:

The reviewers collectively found your submission to be "an interesting manuscript with provocative ideas," quoting one of the reviewers from our consultation session. I concur. Each of the three individual reviews is thoughtful and excellent; I'm appending these in their entirety to this email as I thought that you would want to see them to consider the detailed comments in full as you prepare a revision.

There was one overarching, consistent issue raised in slightly different forms by all reviewers: the under-realization of the ultimate conclusive potential of the paper – at least in its current attempt to answer a quantitative question with non-quantitative means – without more quantitative / formal statistical / modeling development. We offer you the choice of two options: (A) revise the current manuscript to become a shorter perspective/ hypothesis piece detailing the question and what would be needed to answer it (including addressing the reviewer concerns but leaving out a quantitative analysis), or (B) develop a revision with a more comprehensive quantitative/ modeling analysis along the lines suggested by the reviewers.

We're split (in a healthy, constructive way) in terms of slight preference for the two options. At least one reviewer leans towards option A. I myself probably lean towards option B. But we all agree that we would be supportive of your choice of either option so we happily defer to your decision on the ultimate form that you would like for your (promising!) manuscript.

*Reviewer #1 (Recommendations for the authors):*

1. I would like to see formal statistical analyses to support these statements:

• Drug resistance evolution occurs at a surprisingly constant rate (Figure 2A).

How has this been assessed? I would like to see a quantitative assessment of this. The figure title also says "…after the first few years of treatment". What's few here, 2, 4, 10? The rate does not appear visually constant to me over the 10 year span, especially for PI and NRTI.

• While some patients failed without any detectable drug resistance (likely due to non-adherence), viral populations in most patients harbored one, two, three or more resistance mutations.

It does look like the 0 class represents a larger proportion of the area in Figure 2B, but is it a significantly larger proportion?

• The singly-resistant viral populations almost all carry a 3TC mutation, suggesting that resistance to 3TC overwhelmingly occurs before resistance to either the other NRTI or the PI.

I'm not sure that all things having a 3TC mutation is a sufficient reason to say that mutation occurred first (though it is necessary). A different mutation may have occurred first, and then been subsequently outcompeted through clonal interference.

2. I would also please request any revised versions include line numbering to assist in preparing the reviewer's report.

*Reviewer #2 (Recommendations for the authors):*

The manuscript by Feder and colleagues investigates multi-drug resistance in HIV. By analyzing previously published data, they argue that (1) resistance evolves at a constant rate following a transient period of adaptation, (2) resistance evolves via sequences of single mutations, and (3) the order of mutations is largely predictable. The authors argue that these trends cannot be explained by differences in mutation rate and target size, are sometimes at odds with known properties of the corresponding mono-therapies or known properties (e.g. fitness costs) of the known mutants, and are surprising given that the conceptual basis for combination therapies suggests that partially resistant variants should not expand in the population (full 3-drug resistance is required). The authors argue that spatial or temporal heterogeneity in drug concentration could explain these trends.

The paper addresses an important and difficult problem, and it synthesizes a great deal of work from both the clinic and the modeling worlds. The authors make an admirable effort to re-evaluate numerous data sets from a new perspective, and their observations (1-3, above) are important and (in my understanding) under-appreciated findings with potentially important ramifications for understanding how resistance evolves in HIV.

My primary criticism of the paper, as presented, is that the empirical results (1-3) are relatively de-emphasized, and the focus (at least in title and abstract) seems to be on spatial and temporal heterogeneity as an explanation for these results. And while I do find the author's arguments to be plausible – and they nicely outline the qualitative evolutionary features that could arise from spatial and temporal heterogeneity, in part by summarizing previous work from multiple groups – I believe a more quantitative comparison between models and the data would substantially strengthen their conclusions. The paper presents interesting hypotheses and sound qualitative arguments-and this would be ideal for a perspective paper, for example. On the other hand, I think a more detailed analysis of the models would be required to convincingly argue that these heterogeneities are likely, rather than just plausible, explanations for their observations, and to more rigorously determine how features of the heterogeneity lead to particular features of the evolution.

With that said, I recognize that modeling these real-life processes is exceedingly difficult and that the empirical work alone is a nice contribution, and I'm not suggesting that a detailed, realistic model is required. But if the authors prefer to keep the focus on heterogeneities as a likely explanation, I would suggest going in one of two directions (both of which are in line with their expertise and strengths in modeling). One option is to develop a parameterized model based on known properties of the drugs in question (perhaps focusing on just one of the most detailed clinical studies), known mutations rates, etc, and to quantitatively assess the level of heterogeneity needed to (approximately) capture the measured dynamics. It would be interesting to see whether those bounds are consistent with known properties-half life, drug distribution-of the individual drugs and whether, for example, temporal or spatial heterogeneity is the more likely explanation.

Alternatively, I might suggest focusing on a generic toy model-similar to what was likely used to generate the results in Figure 4c – and try to determine qualitative features required for particular features to emerge. In some sense, this approach would be in between what is currently presented (a qualitative model with no equations) and a detailed quantitative model (which may well be impossible given the uncertainties in various parameters). Yet it could provide richer insight into how certain evolutionary outcomes depend on features of the model. In either case, the authors could use these models to make a stronger case for precisely how spatial and temporal heterogeneity (or perhaps a restricted class of such heterogeneity) could lead to the observed phenomena. In particular, it would be interesting to determine the conditions under which the probability of acquiring a mutation rises linearly, or conditions that lead to particular drug ordering (given some set of predefined properties of the mono-therapies themselves). This analysis may substantially limit the structure of the models and would, in my view, enrich the paper a great deal.

I have a few other comments (some naïve) for the authors to consider.

Given that the observed evolutionary dynamics are sometimes unexpected given what's known about the single drug characteristics, what role might interactions between the drugs play? As the authors know, drug interactions have been shown to modulate resistance evolution in other microbial systems (e.g. bacteria). What, if anything, is known about the impacts of drug interactions in HIV? Could this provide an alternative explanation, even in the absence of spatial or temporal heterogeneity?

Similarly, could any of these results be explained by cross resistance / collateral sensitivity. I'm not an expert in viral dynamics, but I know these features have been studied a great deal in bacteria. Is it well established that the mutations found here – which are assumed to be associated with resistance to a single drug – have no collateral effects (i.e. confer increases resistance or sensitivity to more than the target drug)? The authors mention cross-resistance briefly, but it might be worth discussing in a bit more detail.

*Reviewer #3 (Recommendations for the authors):*

Introduction

1. I enjoyed reading the introduction, but I don't think it did the study credit. I would suggest clarifying some of the core points. Specifically, (i) what is missing from previous models (e.g. calibration to data? calibration to specific types of data? processes that are known to occur? inability to recapitulate known phenomena?). I was left a little in the dark at the end of the introduction as to what the study was assessing and how these were improvements as to what had gone before. For example, mention is made of 'clinical data' and 'heterogeneity of selection pressures' but without context, it is unclear what these refer to. I've provided some more specific suggestions below.

2. The introduction is very nice, but I would perhaps be a bit more specific about the nature of 'time' and 'space' as mentioned in the sentence:

"Second, mathematical models have explored how heterogeneity of drug levels in time (Rong et al., 2007; Braithwaite et al., 2006; Rosenbloom et al., 2012; Cadosch et al., 2012; Bershteyn and Eckhoff, 2013) and space (Kepler and Perelson, 1998; Moreno-Gamez et al., 2015; Sanche et al., 2017) can allow the evolution of resistance,…"

as it reads to me a little vague. E.g. what is 'heterogeneity of drug levels'? are the timescales and the spatial scales that have been evaluated? More information would be helpful please.

3. Would remove the word 'quite' as it is ambiguous (same comments applies elsewhere); and perhaps 'Sophisticated' might be replaced by 'complex' or the clause left out entirely.

4. As above (in 1.) "… there is no common framework to test the effects of spatial and temporal heterogeneity on triple-drug combinations of antiretrovirals." is a little vague to me. Can you pin down what you mean by 'spatial and temporal heterogeneity'

5. "…as revealed by various facets of clinical trial and cohort study data" is also vague. Can you explain what these facets are?

6. "Matching models specifically to mutational patterns in clinical data can help us better understand whether these models can explain multi-drug resistance evolution in the real world." – has this been done ? (i.e. calibrating models to mutational patterns within individuals)

7. "…to match clinical data across multiple dimensions" – again, please clarify what you mean by 'multiple dimensions'.

Figure 1

8. Very nice graphic. Could you indicate the timescales of the studies and/or accrual of mutations in the caption?

Section 2

9. "(2) Viruses do not need to be fully resistant to all drugs in a combination in order to spread" – I find this conclusion a little confusing, presumably you mean emerge to detectable levels within a single individual?

Figure 2

10. I'm not convinced by the description of panel A. This in part may be down to more explanation needed of the data. What does the x-axis represent – time since study commencement or time since beginning of therapy? Did all patients keep receiving the triple Tx continually through the study?

11. You conclude that the probability of acquiring a DRM is approximately constant (because the gradients of each of the lines are approximately constant?). However, because the frequencies of DRMs are plotted across the whole population, it's not clear whether the accrual of each DRM is independent or not. Equally, to me it looks like there is saturation of the frequency of these DRMs as well as some more complex cumulative distribution happening. You don't rule these out, but how would these phenomena change your conclusions?

Figure 3B

12. Could the authors give more evidence for the existence of the idealised Hill curve for the drug response? Specifically, at high doses, why would we expect the replication capacity of the drug resistant mutant to be approaching 0?

13. Leaving aside my reservations about the shape of the graph, I believe this graph only captures part of the story; that is, the replication capacity of the mutant existence. Two other factors come into play if a composite measure of 'Chance of mutant emergence' is considered: (i) the probability that the mutant exists (this is a function of the viral load, which itself is a function of R_0_ – i.e. high VL when R_0_>1, low/negligible VL when R_0_<1) and (ii) the probability of an outbreak for a given R_0_ and existence of a mutant. Approximately this probability is 1-1/R_0_, then finally (iii) the relative replication capacity vs wild type. The chance of mutant emergence would then be the product of all three. This is just an idea, and by no means a suggestions for inclusion, but it's been the way I think about emergence of drug resistance and may be useful for these purposes.

---

## [Author Response]

Essential revisions:The reviewers collectively found your submission to be "an interesting manuscript with provocative ideas," quoting one of the reviewers from our consultation session. I concur. Each of the three individual reviews is thoughtful and excellent; I'm appending these in their entirety to this email as I thought that you would want to see them to consider the detailed comments in full as you prepare a revision.There was one overarching, consistent issue raised in slightly different forms by all reviewers: the under-realization of the ultimate conclusive potential of the paper – at least in its current attempt to answer a quantitative question with non-quantitative means – without more quantitative / formal statistical / modeling development. We offer you the choice of two options: (A) revise the current manuscript to become a shorter perspective/ hypothesis piece detailing the question and what would be needed to answer it (including addressing the reviewer concerns but leaving out a quantitative analysis), or (B) develop a revision with a more comprehensive quantitative/ modeling analysis along the lines suggested by the reviewers.We're split (in a healthy, constructive way) in terms of slight preference for the two options. At least one reviewer leans towards option A. I myself probably lean towards option B. But we all agree that we would be supportive of your choice of either option so we happily defer to your decision on the ultimate form that you would like for your (promising!) manuscript.

Thank you for the thoughtful reviews and the positive assessment of our work. In response to your suggestions, we have added several new quantitative analyses that we believe have improved the manuscript. These major changes fall into two main categories:

First, we quantified patterns displayed in Figure 2 corresponding to the major clinical observations concerning ongoing, sequential and partially predictable evolution of drug resistance in individuals treated with combination therapy. For the results in 2A (ongoing resistance evolution) we worked with Dr Chanson Brumme, one of the authors of the BC study that showed an approximately linear increase in resistance over time. Because of this new collaboration, we are now able to quantify the yearly probability of resistance evolution each year after the start of therapy (conditional on no resistance to this point) – which makes it clear that while resistance rates do decline over time, this decline is relatively slow and significant ongoing resistance does emerge even after years of successful virologic suppression. For the results in 2C (partially predictable order of mutations), we created quantitative null expectations about which mutation is expected to occur first based on mutation rates which allowed us to formally test whether our observations (that 3TC resistance occurs first on PI-based regimens and NNRTI resistance occurs first on NNRTI-based regimens) can reject this null model. We find that in 14 out of 15 treatment regimes, we can indeed reject the null model.

Second, we extended two pre-existing models of spatial (Moreno-Gamez et al. 2015) or temporal (Rosenbloom et al. 2012) heterogeneity of drug levels to demonstrate quantitatively which clinical observations could or could not be derived from these models. Following reviewer 2’s suggestion that we focus on a particular therapy where parameters were relatively well described, we elected to investigate 3TC+D4T+NFV, a protease inhibitor based therapy that we originally used to introduce the patterns in Figure 1A, for which we had relatively complete information about mutation selection windows, resistance profiles and drug penetrance. We find that under these two models, some – but not all – of the clinical attributes emerge under both spatial and temporal heterogeneity (plotted in the new Figure 5). These results provide a proof of concept that matching multiple quantitative attributes of clinical data to models can help us understand which modeled features are insufficiently complex, and lay the groundwork for a much more extensive investigation into factors like more realistic drug adherence and penetrance profiles. In the initial submission, we argued for the need for this type of modeling, closely linked to clinical data and comparative across a wide range of triple-drug combinations. Including an investigation into a particular therapy has made the need for this type of work even clearer.

Finally, we have embraced the reviewers’ suggestions for clarity, and believe the manuscript is now more understandable, in addition to being more quantitative.

Reviewer #1 (Recommendations for the authors):1. I would like to see formal statistical analyses to support these statements:• Drug resistance evolution occurs at a surprisingly constant rate (Figure 2A).How has this been assessed? I would like to see a quantitative assessment of this. The figure title also says "…after the first few years of treatment". What's few here, 2, 4, 10? The rate does not appear visually constant to me over the 10 year span, especially for PI and NRTI.

The data available to us at the time of initial submission prevented us from computing these statistics. To overcome this gap, we have brought in Chanson Brumme, an author on the original study measuring and reporting the accumulation of resistance over time. With the assistance of Dr. Brumme, we have been able to compute the conditional probability of resistance in any given year of treatment using a survival analysis approach (see new Figure 2A). As the reviewer notes, interpreting these cumulative curves is not straightforward, and stratifying by the yearly probability of resistance evolution circumvents some difficulties related to a decreasing sample size as more individuals become resistant.

In computing these probabilities, it became clear that while the probability of resistance evolution is highest in the first few (2-3) years of therapy as we previously reported, the probability of resistance evolution does not stabilize to a constant rate for each of the four drug resistance mutation classes (3TC/FTC, other NRTIs, NNRTIs and PIs). However, the rate decreases sufficiently slowly that overall the pattern appears approximately linear in the original paper (Rocheleau et al. 2018). Because of our increased ability to measure the yearly rates, we now do not believe that the rates are entirely constant but rather are decreasing but non-zero for several classes of drugs for at least ten years after therapy, and are well-fit by an exponential decay. We later explore the reason for the decrease in rate in our newly added modeling section and hypothesize that it may emerge because of differences in viral setpoints across patients.

• While some patients failed without any detectable drug resistance (likely due to non-adherence), viral populations in most patients harbored one, two, three or more resistance mutations.It does look like the 0 class represents a larger proportion of the area in Figure 2B, but is it a significantly larger proportion?

We did not intend this as a claim, as the relative amounts of 0, 1, 2, and 3 resistance mutation is in part a function of the sample characteristics. We’re reading very little into the percentage of patients failing without any resistance as opposed to the number of patients failing with a single resistance mutation. We’ve clarified the language in the text to read that viral populations in *many* populations harbored one, two, three or more resistance mutations.

• The singly-resistant viral populations almost all carry a 3TC mutation, suggesting that resistance to 3TC overwhelmingly occurs before resistance to either the other NRTI or the PI.

We have added a statistical analysis to test our two assertions: (1) On NNRTI-based therapies, NNRTI resistance occurs first and (2) on PI-based therapies, 3TC/FTC resistance occurs first. See the methods for full details, but briefly, for each therapy consisting of three drugs (except 3TC+ABC+LPV, see below), we calculated every single-step nucleotide substitution that could create a drug resistance mutation to any of the three categories of drugs. For each of those mutations, we took the mutation rate corresponding to the nucleotide substitution (i.e., A → T, T → A, etc. ) from the published literature. We summed those mutation rates across all possible drug resistance mutation-causing nucleotide substitutions to compute, separately for each drug, the rate at which resistance to such a drug should arise. For NNRTI-based therapies, we can then compute the relative probability that NNRTI resistance should emerge first, versus either of the other two categories (NRTI or 3TC/FTC). For PI-based therapies, we can compute the relative probability that PI resistance should emerge first, versus either of the other two categories (other NRTI or PI). Using these probabilities, we can compute a one-sided binomial test on these two hypotheses (NNRTI mutations precede NRTI or 3TC/FTC mutations or 3TC/FTC mutations precede other NRTI or PI mutations) separately for each three drug combination using the bespoke probabilities for each drug combination. The number of successes is the number of times the tested category appears first and the total number of trials are the number of single-DRM mutations observed on a given therapy. The full data to compute all the tests, their p-values and the comparison with a Benjamini-Hochberg correction is given in supplemental tables 5 and 6, and significance values have now been added to Figure 2C.

In all cases except one with quite small sample size (only 15 observations), these tests support rejecting the null hypothesis that first mutations occur proportionally to what we might expect based on mutation rates: in the case of NNRTI-based therapies, NNRTI mutations occur at a higher rate than expected. In the case of PI-based therapies, 3TC/FTC mutations occur at a higher rate than expected.

I'm not sure that all things having a 3TC mutation is a sufficient reason to say that mutation occurred first (though it is necessary). A different mutation may have occurred first, and then been subsequently outcompeted through clonal interference.

This point is well-taken and we have added a caveat in the text:

“It is also possible that other classes of resistance mutations rise to high frequency before the first mutations described above but are then outcompeted due to clonal interference, but consistently, the first resistance mutations we see belong to a single category. (lines 200-203)”.

2. I would also please request any revised versions include line numbering to assist in preparing the reviewer's report.

This has been fixed.

Reviewer #2 (Recommendations for the authors):The manuscript by Feder and colleagues investigates multi-drug resistance in HIV. By analyzing previously published data, they argue that (1) resistance evolves at a constant rate following a transient period of adaptation, (2) resistance evolves via sequences of single mutations, and (3) the order of mutations is largely predictable. The authors argue that these trends cannot be explained by differences in mutation rate and target size, are sometimes at odds with known properties of the corresponding mono-therapies or known properties (e.g. fitness costs) of the known mutants, and are surprising given that the conceptual basis for combination therapies suggests that partially resistant variants should not expand in the population (full 3-drug resistance is required). The authors argue that spatial or temporal heterogeneity in drug concentration could explain these trends.The paper addresses an important and difficult problem, and it synthesizes a great deal of work from both the clinic and the modeling worlds. The authors make an admirable effort to re-evaluate numerous data sets from a new perspective, and their observations (1-3, above) are important and (in my understanding) under-appreciated findings with potentially important ramifications for understanding how resistance evolves in HIV.My primary criticism of the paper, as presented, is that the empirical results (1-3) are relatively de-emphasized, and the focus (at least in title and abstract) seems to be on spatial and temporal heterogeneity as an explanation for these results. And while I do find the author's arguments to be plausible – and they nicely outline the qualitative evolutionary features that could arise from spatial and temporal heterogeneity, in part by summarizing previous work from multiple groups – I believe a more quantitative comparison between models and the data would substantially strengthen their conclusions. The paper presents interesting hypotheses and sound qualitative arguments – and this would be ideal for a perspective paper, for example. On the other hand, I think a more detailed analysis of the models would be required to convincingly argue that these heterogeneities are likely, rather than just plausible, explanations for their observations, and to more rigorously determine how features of the heterogeneity lead to particular features of the evolution.With that said, I recognize that modeling these real-life processes is exceedingly difficult and that the empirical work alone is a nice contribution, and I'm not suggesting that a detailed, realistic model is required. But if the authors prefer to keep the focus on heterogeneities as a likely explanation, I would suggest going in one of two directions (both of which are in line with their expertise and strengths in modeling). One option is to develop a parameterized model based on known properties of the drugs in question (perhaps focusing on just one of the most detailed clinical studies), known mutations rates, etc, and to quantitatively assess the level of heterogeneity needed to (approximately) capture the measured dynamics. It would be interesting to see whether those bounds are consistent with known properties – half life, drug distribution – of the individual drugs and whether, for example, temporal or spatial heterogeneity is the more likely explanation.Alternatively, I might suggest focusing on a generic toy model – similar to what was likely used to generate the results in Figure 4c – and try to determine qualitative features required for particular features to emerge. In some sense, this approach would be in between what is currently presented (a qualitative model with no equations) and a detailed quantitative model (which may well be impossible given the uncertainties in various parameters). Yet it could provide richer insight into how certain evolutionary outcomes depend on features of the model. In either case, the authors could use these models to make a stronger case for precisely how spatial and temporal heterogeneity (or perhaps a restricted class of such heterogeneity) could lead to the observed phenomena. In particular, it would be interesting to determine the conditions under which the probability of acquiring a mutation rises linearly, or conditions that lead to particular drug ordering (given some set of predefined properties of the mono-therapies themselves). This analysis may substantially limit the structure of the models and would, in my view, enrich the paper a great deal.

In general, we completely agree with Reviewer #2 (and Reviewer #3) about the potential added value of supplementing the paper with some basic modeling. We thought carefully about Reviewer #2’s two suggestions and decided that their first idea (suggestion #1) would add more substantively to the discussion. To some extent, these toy models (suggestion #2) already exist in the literature, although more often with two drugs than with three. While these toy models can and should be examined for their tendencies to create the patterns observed in clinical data, we believe that matching these models as closely as possible to measured HIV parameters is most useful in understanding how drug resistance evolves in triple-drug treated HIV populations. Similar to what Reviewer #2 suggested, we focused on a combination of drugs with well-characterized mutation rates, mutant-selection windows, drug penetrances across multiple compartments, half-life and detailed clinical data (i.e., what is plotted in Figure 1A and Figure 2B and C) – 3TC+D4T+NFV. We extended two existing models of of spatial (Moreno-Gamez et al. 2015) or temporal (Rosenbloom et al. 2012) heterogeneity (via incomplete drug penetrance or adherence, respectively) to account for three drugs and simulated 1500 patients where we examined clinical features (resistance timing, number of mutations and order of mutations) similar to our analysis on real viral data. In doing so, we discovered that, for example, while the model of temporal heterogeneity can create sequential and predictable evolution of resistance, under such a model, very little resistance evolution emerges after initial virologic suppression, even in patients with moderate or low adherence. This model outcome is inconsistent with the ongoing resistance evolution observed so frequently in individuals with HIV. This finding validates Reviewer #2’s concern (and indeed our argument in the initial submission) that quantitative models paired with clinical data are necessary to understand the evolution of multi-drug resistance, and motivates new questions about which types of adherence behaviors can allow ongoing resistance to emerge.

While we still very much believe that a future study should compare patterns across many different types of triple drug therapies, starting with one well-characterized therapy already helps us understand which clinical patterns emerge straightforwardly from simple models and which ones do not, and motivates future thinking about how these models must be extended.

I have a few other comments (some naïve) for the authors to consider.Given that the observed evolutionary dynamics are sometimes unexpected given what's known about the single drug characteristics, what role might interactions between the drugs play? As the authors know, drug interactions have been shown to modulate resistance evolution in other microbial systems (e.g. bacteria). What, if anything, is known about the impacts of drug interactions in HIV? Could this provide an alternative explanation, even in the absence of spatial or temporal heterogeneity?

This question has been investigated in vitro by measuring dose response response curves under pairwise combinations of drugs, and has revealed that not all drugs work additively. The modeling in that paper suggests that some of these non-additive interactions do correlate with rates of viremia in clinical trial data. We’ve added the following note to the forward-looking discussion as an important point that future studies will need to grapple with:

“Experimental work has highlighted drug pairs that suppress in vitro viral populations to both greater and lesser extents than each drug working independently, which may provide a partial answer to why some combinations of drugs fail at a surprisingly high rate (Jilek et al. 2012). (lines 475-477)”.

In our own new modeling section, we used data from Jilek et al. 2012 to choose how drugs should combine across drug classes.

Similarly, could any of these results be explained by cross resistance / collateral sensitivity. I'm not an expert in viral dynamics, but I know these features have been studied a great deal in bacteria. Is it well established that the mutations found here-which are assumed to be associated with resistance to a single drug – have no collateral effects (i.e. confer increases resistance or sensitivity to more than the target drug)? The authors mention cross-resistance briefly, but it might be worth discussing in a bit more detail.

In HIV, there are no single mutations (for example, ones that upregulate efflux pumps) that confer strong resistance to multiple classes of drugs (although there are mutations that provide cross resistance within a class, for example, NRTIs). This comment has highlighted that we need to state this more explicitly in the text in two ways:

– We’ve added the following language in the introduction: “Because no mutations confer resistance across multiple classes of anti-retrovirals (unlike many cases of bacterial resistance), even if a virus gets a mutation conferring resistance to one drug, the two other drugs keep the R_0_ of that virus below one, and so viruses carrying this mutation cannot increase in number” (lines 31-32).

– In a small number of instances, major 3TC/FTC mutations also confer resistance to other NRTIs (namely ABC), which makes interpretation of these results less straightforward, because a single mutation confers some degree of resistance to multiple drugs. There are two 3TC+ABC backbone therapies examined here (+ a PI – LPV, n = 11, and + an NNRTI – EFV, n = 16). We have added caveats about 3TC/ABC combination therapies to the discussion of the which DRM comes first analysis in 2C. (lines 191-193).

Reviewer #3 (Recommendations for the authors):Introduction1. I enjoyed reading the introduction, but I don't think it did the study credit. I would suggest clarifying some of the core points. Specifically, (i) what is missing from previous models (e.g. calibration to data? calibration to specific types of data? processes that are known to occur? inability to recapitulate known phenomena?). I was left a little in the dark at the end of the introduction as to what the study was assessing and how these were improvements as to what had gone before. For example, mention is made of 'clinical data' and 'heterogeneity of selection pressures' but without context, it is unclear what these refer to. I've provided some more specific suggestions below.

On re-reading the introduction, we agree that many aspects of how we describe space and time are not clear as written. We hope these descriptions become clear as the paper progresses, but the reviewer is right that these definitions and shortcomings need to be made more clear upfront. Individual comments addressed below.

2. The introduction is very nice, but I would perhaps be a bit more specific about the nature of 'time' and 'space' as mentioned in the sentence:"Second, mathematical models have explored how heterogeneity of drug levels in time (Rong et al., 2007; Braithwaite et al., 2006; Rosenbloom et al., 2012; Cadosch et al., 2012; Bershteyn and Eckhoff, 2013) and space (Kepler and Perelson, 1998; Moreno-Gamez et al., 2015; Sanche et al., 2017) can allow the evolution of resistance,…"as it reads to me a little vague. E.g. what is 'heterogeneity of drug levels'? are the timescales and the spatial scales that have been evaluated? More information would be helpful please.

We have edited the language surrounding our first usage of time and space in this sentence to give much more concrete examples of what we mean by heterogeneity in space and time. It now reads as follows:

“Second, mathematical models have explored how heterogeneous drug levels within a patient over time (for example, due to drug metabolism between doses or treatment interruptions within single patients) (Rong et al., 2007; Braithwaite et al., 2006; Rosenbloom et al., 2012; Cadosch et al., 2012; Bershteyn and Eckhoff, 2013) and space (for example, due to variable drug penetrance to different anatomical sites within a patient) (Kepler and Perelson, 1998; Moreno-Gamez et al., 2015; Sanche et al., 2017) can allow the evolution of resistance…” (lines 47-50).

3. Would remove the word 'quite' as it is ambiguous (same comments applies elsewhere); and perhaps 'Sophisticated' might be replaced by 'complex' or the clause left out entirely.

We have made this suggested change.

4. As above (in 1.) "… there is no common framework to test the effects of spatial and temporal heterogeneity on triple-drug combinations of antiretrovirals." is a little vague to me. Can you pin down what you mean by 'spatial and temporal heterogeneity'

We have improved the clarity of the language here. The sentence now reads as follows:

“…there is no common framework to test how spatial and temporal variation of drug levels in the body contributes to HIV multi-drug resistance evolution on triple-drug therapies.” (lines 53-56).

5. "…as revealed by various facets of clinical trial and cohort study data" is also vague. Can you explain what these facets are?

To help clarify, we added language to the following sentence so it now reads as follows:

“Specifically, matching models to mutational patterns in clinical data (i.e., which mutations happen, how often, on what timescales, and in what order) can help us better understand whether these models can explain multi-drug resistance evolution in the real world.” (lines 59-60).

6. "Matching models specifically to mutational patterns in clinical data can help us better understand whether these models can explain multi-drug resistance evolution in the real world." – has this been done (i.e. calibrating models to mutational patterns within individuals)?

In the interest of attempting to not lengthen the introduction too much, we elected to summarize major studies on the evolution of HIV drug resistance due to spatial or temporal variability in drug levels in Supplemental Table 1. We hope this better describes why no existing studies manage to address this important problem.

7. "…to match clinical data across multiple dimensions" – again, please clarify what you mean by 'multiple dimensions'

We added some clarifying language. The end of the sentence now reads:

“We close by suggesting a path forward in expanding promising mathematical models of HIV to understand three-drug combinations and testing the models’ ability to match clinical data across multiple dimensions – for example, not just how often drugs fail, but the types of mutational patterns these failures leave in viral genetic data.” (lines 71-72).

Figure 18. Very nice graphic. Could you indicate the timescales of the studies and/or accrual of mutations in the caption?

We have added these timescales to the figure caption.

Section 29. "(2) Viruses do not need to be fully resistant to all drugs in a combination in order to spread" – I find this conclusion a little confusing, presumably you mean emerge to detectable levels within a single individual?

This is correct. We have updated the language here:

“Viruses do not need to be fully resistant to all drugs in a combination in order to spread to detectable levels within individual patients.” (line 115)

Figure 210. I'm not convinced by the description of panel A. This in part may be down to more explanation needed of the data. What does the x-axis represent – time since study commencement or time since beginning of therapy? Did all patients keep receiving the triple Tx continually through the study?

We have replaced Figure 2A with a panel that hopefully captures the same information more clearly. In answer to the reviewer’s questions, time represents the amount of time since the patient started on triple-drug therapy. Patients are removed from the dataset when they are first observed to have a resistance mutation.

Figure 2A now fits the yearly probability of resistance evolution (conditional on no resistance evolution up to this point) at set numbers of years after therapy initiation.

11. You conclude that the probability of acquiring a DRM is approximately constant (because the gradients of each of the lines are approximately constant?). However, because the frequencies of DRMs are plotted across the whole population, it's not clear whether the accrual of each DRM is independent or not. Equally, to me it looks like there is saturation of the frequency of these DRMs as well as some more complex cumulative distribution happening. You don't rule these out, but how would these phenomena change your conclusions?

Again, we agree with Reviewer #3 (and Reviewer #1) that these cumulative distributions are challenging to interpret, and that the change in the overall population size impacts the appearance of the curves. Because of this, we have switched to looking at the probability of resistance evolution given no resistance up to this point in time (i.e., a conditional survival analysis). Please see our response to Reviewer #1, but to address specific points raised here,

1. Re: independence: Given the data that we have access to, it’s not possible to determine if resistance mutations are independent, but based on evidence from Figure 2C (which is a different dataset), it is probable that they are not. These individuals are given combinations of drugs (for example, a PI, 3TC, and a different NRTI). If an individual given these three drugs acquires 3TC resistance, they are censored on the PI and NRTI curves. This is standard in survival analysis, and removes observations from the denominator of the survival probability. This does not inflate the probability or resistance evolution for the censored drug class, and incorporates information that an individual survived some number of years without any resistance evolution for a certain class of drugs (even if they ultimately became resistant to a different drug).

2. Re: saturation. Access to the raw data underlying the initial version of Figure 2A has revealed that the rates of resistance evolution do appear to decrease over time. We now plot the conditional survival (resistance evolution) probability, so it does not appear that the saturation emerges because the number of individuals becomes too small (i.e., no more resistance can evolve because everyone has become resistant). However, some of our spatial modeling results suggest that saturation may be affecting the shape of this curve if initially all individuals with higher than average viral population sizes evolve resistance early, and as time goes on, only individuals with increasingly small population sizes remain. We’ve added discussion of this possibility to the text. Please also see the comment in response to Reviewer #1.

Figure 3B12. Could the authors give more evidence for the existence of the idealised Hill curve for the drug response? Specifically, at high doses, why would we expect the replication capacity of the drug resistant mutant to be approaching 0?

These curves have been measured empirically in in vitro settings. We thank the reviewer for pointing out that this was not clear as written, and we have added two sentences describing the existence of these experiments.

“Such curves have been measured in cell culture for anti-retroviral-treated HIV (Sampah et al., 2011).” (lines 229-230) and “Drugs overpowering even resistant virus at high concentrations has been demonstrated empirically in experimental settings (Sampah et al., 2011).” (lines 239-241).

13. Leaving aside my reservations about the shape of the graph, I believe this graph only captures part of the story; that is, the replication capacity of the mutant existence. Two other factors come into play if a composite measure of 'Chance of mutant emergence' is considered: (i) the probability that the mutant exists (this is a function of the viral load, which itself is a function of R_0_ – i.e. high VL when R_0_>1, low/negligible VL when R_0_<1) and (ii) the probability of an outbreak for a given R_0_ and existence of a mutant. Approximately this probability is 1-1/R_0_, then finally (iii) the relative replication capacity vs wild type. The chance of mutant emergence would then be the product of all three. This is just an idea, and by no means a suggestions for inclusion, but it's been the way I think about emergence of drug resistance and may be useful for these purposes.

We entirely agree – in our opinion, especially the first point about population size is of critical importance. We had previously discussed this point in the section about “covering the tail” of long half-life NNRTIs at therapy cessation with shorter half-life drugs (so that an NNRTI would not exist as the only therapy during its decay). Studies that have examined this hypothesis found that covering the tail did not change the rate of drug resistance emergence, presumably because the population size is too small to generate the relevant NNRTI resistance mutations. We extend this discussion in the new version of the manuscript, because it directly impacts our finding that even under simple models of incomplete adherence causing temporal heterogeneity of drug levels, resistance evolution does not evolve after initial suppression.